# From "Sure" to "Sorry": Detecting Jailbreak in Large Vision Language Model via JailNeurons

**Yuyou Gan[1], Qingming Li[1], Junhao Li[2], Zhi Chen[3], Jinbao Li[4], Xiaoming Li[5]\*, Shouling Ji[1,6]\***

[1]Zhejiang University, [2]Guangzhou University, [3]University of Illinois at Urbana-Champaign,
[4]Qilu University of Technology, [5]Zhejiang Yuexiu University
[6]Zhejiang Key Laboratory of Decision Intelligence
ganyuyou@zju.edu.cn, lxm696@tju.edu.cn, sji@zju.edu.cn

## Abstract

Large Vision-Language Models (LVLMs) are vulnerable to jailbreak attacks that can generate harmful content. Existing detection methods are either limited to detecting specific attack types or are too time-consuming, making them impractical for real-world deployment. To address these challenges, we propose **JDJN** (**J**ailbreak **D**etection via **J**ail**N**eurons), a novel jailbreak detection method for LVLMs. Specifically, we focus on **JailNeurons**, which are key neurons related to jailbreak at each model layer. Unlike the "SafeNeurons", which explain why aligned models can reject ordinary harmful queries, JailNeurons capture how jailbreak prompts circumvent safety mechanisms. They provide an important and previously underexplored complement to existing safety research. We design a neuron localization algorithm to detect these JailNeurons and then aggregate them across layers to train a generalizable detector. Experimental results demonstrate that our method effectively extracts jailbreak-related information from high-dimensional hidden states. As a result, our approach achieves the highest detection success rate with exceptionally low false positive rates. Furthermore, the detector exhibits strong generalizability, maintaining high detection success rates across unseen benign datasets and attack types. Finally, our method is computationally efficient, with low training costs and fast inference speeds, highlighting its potential for real-world deployment.

## 1 Introduction

Large Vision-Language Models (LVLMs) exhibit impressive vision-language capabilities and have consequently become a focal point of research in both industry and academia Wang et al. (2024b); Zhu et al. (2023); Liu et al. (2023a). While LVLMs inherit the powerful language capabilities of LLMs, they also amplify the associated security risks Carlini et al. (2023). Among these risks, jailbreak attacks pose a significant threat, wherein an adversary adversarially crafts inputs to compel the model to generate harmful or prohibited content. The inclusion of the visual modality expands the attack surface, enabling more diverse and sophisticated jailbreak methods that are consequently harder to defend against. In contrast to text-only LLMs, attacks on LVLMs can exploit the interplay between visual and textual inputs. These attacks primarily fall into three categories, as illustrated in Figure 1: (i) Injecting adversarial perturbations into images via gradient-based optimization to elicit specific malicious outputs Carlini et al. (2023); Yin et al. (2023); Zhao et al. (2023). (ii) Embedding malicious text into images as rendered characters to bypass the model's security mechanisms Gong et al. (2025). (iii) Selecting images semantically correlated with harmful concepts to pair with text, thereby increasing the maliciousness of the output Liu et al. (2023b).

To address these threats, most existing defense methods for LVLMs borrow directly from LLM defenses and can be broadly divided into two categories. The first is training-phase defenses, such as

---

*Corresponding Authors

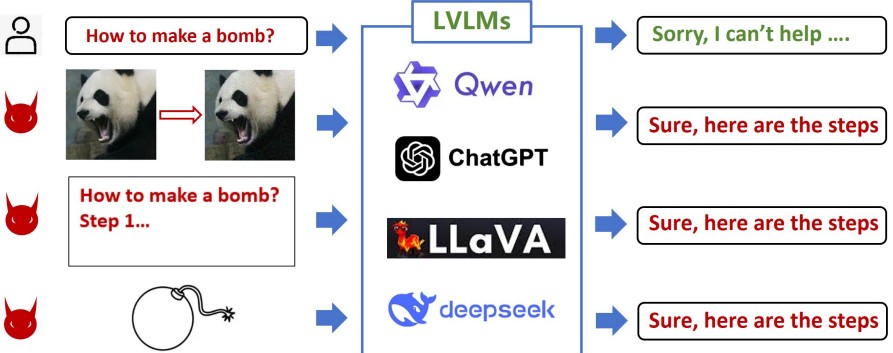

Figure 1: Illustration of three primary types of jailbreak attacks targeting LVLMs.

safety alignment Chen et al. (2024); Li et al. (2024); Zong et al. (2024), which typically incur substantial computational overhead and costly data annotation. The second is inference-phase defenses, such as jailbreak detection via preprocessing the input Xu et al. (2024), evaluating the output Zhang et al. (2023); Gou et al. (2024) and performing semantic checks on the intermediate representations Jiang et al. (2025). However, inference-time defenses often suffer from issues such as increased latency and limited generalization to unseen attack types or benign examples.

In our work, we take a neuron-level perspective by identifying and leveraging abnormal neurons (which we term **JailNeurons**) that are specifically activated by jailbreak inputs. In contrast to the previously studied safety mechanism Wei et al. (2024); Zhou et al. (2024b), which explain how safety mechanisms of aligned models refuse standard harmful queries (which we term **SafeNeurons**), JailNeurons form a distinct set that capture how jailbreak attacks succeed in subverting safety mechanisms. Our method therefore complements the prior work while targeting a novel and largely unexplored field in LVLM security. However, there are two challenges. The first challenge lies in confirming whether JailNeurons truly exist: while prior studies suggest jailbreak-related signals in model representations, they have not localized them to a small set of neurons Zhou et al. (2024a); Jiang et al. (2025). Second, even if these neurons can be identified, how JailNeurons can be exploited for jailbreak detection and whether they generalize to out-of-distribution (OOD) attacks remain open questions.

In this work, we focus on identifying neurons that are specifically associated with jailbreak behaviors, and propose **JDJN** (**J**ailbreak **D**etection via **J**ail**N**eurons), a novel, efficient, and generalizable approach for detecting jailbreak attacks in LVLMs. To address the first challenge, we conduct an empirical investigation of LVLMs under jailbreak attacks and verify that neuron activations triggered by jailbreak inputs are indeed separable from those of benign inputs. Building on this finding, we introduce a **"sure-to-sorry" localization procedure** that progressively narrows down the candidate set of neurons and enables us to pinpoint those most strongly associated with jailbreak behaviors (i.e., JailNeurons). To address the second challenge, we propose a **"top-to-bottom" selection strategy** to select multiple layers. Finally, we aggregate the activations of selected neurons from these key layers and train a lightweight classifier (e.g., an SVM), which yields an efficient and generalizable approach for detecting jailbreak inputs.

We conduct extensive experiments to validate our method's performance across four distinct LVLMs, three different jailbreak attack types, and three benign datasets with varying distributions. The results demonstrate that our method significantly outperforms existing baselines. For instance, on the LLaVA model Liu et al. (2023a), JDJN achieves over 99% true positive rate (TPR) at less than 1% false positive rate (FPR) on seen attack types. Critically, it also shows remarkable generalization, maintaining over 94% TPR at less than 2% FPR on unseen attacks and OOD benign data. Furthermore, JailNeuron is both data-efficient, requiring only a few hundred samples for training, and computationally lightweight. It operates non-intrusively without modifying the target LVLM, imposing negligible inference overhead, which makes it practical for real-time applications. Finally, ablation studies confirm that our neuron localization strategy effectively identifies JailNeurons, outperforming alternative selection methods.

Overall, the core contributions of our work are as follows:

- We provide a systematic analysis demonstrating that jailbreak and benign inputs create distinguishable activation patterns within LVLMs. We show that these discriminative signals are distributed across multiple layers, with different attack types affecting different parts of the model.

- We propose a novel, principled method for identifying JailNeurons by training layer-wise masks. This approach effectively isolates salient signals from high-dimensional noise and mitigates overfitting.

- We introduce JDJN, a lightweight and efficient jailbreak detection framework. Extensive experiments show that JDJN achieves state-of-the-art performance, maintaining a high TPR at a near-zero FPR, and demonstrates remarkable generalization to unseen attacks and OOD data.

## 2 RELATED WORK

**Jailbreak Detection on LVLMs.** Existing methods for jailbreak detection in LVLMs can be broadly categorized into three groups. The first class focuses on input preprocessing, where the sensitivity of the model to transformed inputs is examined to reveal adversarial intent Xu et al. (2024); Zhang et al. (2023). The second class centers on output analysis, including techniques that employ external classifiers to judge harmfulness or prompt the victim model itself to inspect its own responses Gou et al. (2024); Pi et al. (2024). A third line of work investigates abnormal internal activations Jiang et al. (2025). The most relevant to our approach uses a logit lens to extract semantic information from every layer and measures its similarity to predefined refusal fragments, thereby detecting jailbreak samples.

**Security Mechanisms of LLMs and LVLMs.** Security mechanisms of LLMs are explored from two perspectives: (1) High-dimensional representation analysis, examining semantic information in layer representations using tools like Logit lens Belrose et al. (2023) or steering vectors Wang et al. (2024a); Burns et al. (2022); Moschella et al. (2022). (2) Internal structure analysis, identifying secure neurons for fine-tuning, such as using SNIP Wei et al. (2024) to locate key neurons Zhao et al.; He et al. (2024); Wei et al. (2024). To the best of our knowledge, this work is the first to study LVLM jailbreaking mechanisms via neuron activation values and proposes an effective detection algorithm.

## 3 THREAT MODEL

We assume the defender has white-box access to the target model, including its internal activations and parameter gradients. This allows identifying neuron-level behaviors that are correlated with jailbreak phenomena.

The defender can collect a small set of successful jailbreak samples $\mathcal{X}_{j1}$ and a batch of benign samples $\mathcal{X}_{b1}$. Using these, the defender trains a detector that should generalize to unseen distributions: namely, it should achieve high TPR on jailbreak inputs from other distributions $(\mathcal{X}_{j2}, \mathcal{X}_{j3}, \ldots)$ while maintaining extremely low FPR on benign distributions $(\mathcal{X}_{b2}, \mathcal{X}_{b3}, \ldots)$.

This setting is consistent with prior jailbreak detection studies Jiang et al. (2025); Xu et al. (2024), which likewise adopt a white-box assumption to extract features for building robust detectors.

## 4 METHODOLOGY

### 4.1 WARM-UP: DETECT JAILBREAK SAMPLES WITH ONE-LAYER ACTIVATIONS

Our method is inspired by Zhou et al. (2024a), who demonstrated that benign and jailbreak samples in LLMs can be distinguished by applying a linear classifier to neuron activations at each decoder layer, thereby confirming that jailbreak-related information is embedded in internal representations. In contrast, we focus on LVLMs, where this property has not yet been established. Moreover, their work did not investigate the robustness of detectors to OOD attack samples and benign data, which is a crucial aspect for practical jailbreak detection. To address these points, we conduct a preliminary

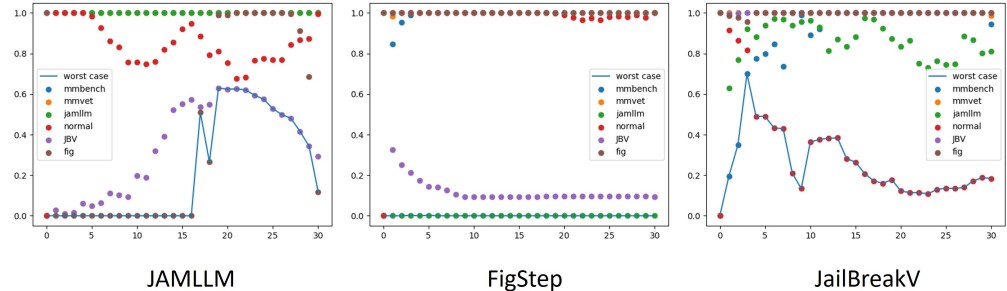

Figure 2: This figure plots detector accuracy against the neuron activation source layer on Janus-pro. Different colors denote test datasets from six distributions, and blue dashed lines indicate the worst-case performance per layer.

study in LVLMs and formulate two guiding research questions: (i) Given a specific attack dataset and a benign dataset, are their hidden state vectors linearly separable? (ii) Can a linear classifier trained on one pair of attack and benign datasets transfer to other types of attacks and benign data?

To answer these questions, we select four state-of-the-art LVLMs: MiniGPT4-7B Zhu et al. (2023), LLaVA-v1.5-7B Liu et al. (2023a), Qwen2-VL-7B Wang et al. (2024b) and Janus-pro-7B Chen et al. (2025). We generate jailbreak samples using three attack methods: JAMLLM Niu et al. (2024), FigStep Gong et al. (2025), and JailBreakV Luo et al. (2024). Our benign data comprises samples from three diverse sources: MM-Bench Liu et al. (2024), MM-Vet Yu et al. (2023), and a set of general-purpose prompts (Normal Prompts) Zhou et al. (2024a).

For the four datasets other than MM-Bench and MM-Vet, we randomly generate or sample 400 instances; for MM-Bench and MM-Vet, we use 200 MM-Bench instances and 218 MM-Vet instances, since these constitute all of their available data. We then extract hidden state vectors from all layers for each of the six data distributions. To evaluate generalization, we treat FigStep, JailBreakV, JAMLLM as known attacks and MM-Vet as a known benign dataset. The remaining two datasets (MM-Bench, Normal Prompts) are held out as unknown test sets. We use a 4/1 split for training and testing on the known datasets.

For each layer, we train three separate SVM classifiers: one on (FigStep, MM-Vet), one on (JailBreakV, MM-Vet), and one on (JAMLLM, MM-Vet). We then evaluate each classifier on both in-distribution (ID) and OOD test sets. The results of Janus-pro are shown in Figure 2. The reults for the other three models are shown in the Appendix A.2. The results lead to two key observations: (i) **Linear Separability.** Consistent with the findings in LLMs, a linear classifier can achieve a high classification accuracy on ID data. Nearly every layer achieves a classification accuracy close to 100% on the ID data. (ii) **Poor Generalization.** No single layer generalizes well to all OOD samples. As the blue dashed line indicates, the worst-case accuracy for any given layer consistently falls below 80%.

## 4.2 JDJN: JAILBREAK DETECTION VIA JAILNEURON

In our preliminary experiment, we train an SVM using the activations from a single layer to distinguish benign from jailbreak samples. While effective on seen attacks, the model fails to generalize to unseen ones. We attribute this to two main factors. (i) The full activation vector from one layer contains substantial jailbreak-irrelevant noise. As suggested by the SafeNeurons study Zhou et al. (2024b); Zhang et al. (2025), only a small fraction of neurons are directly associated with safety, reflecting the sparsity and redundancy of modern language models Frantar & Alistarh (2023); Sun et al. (2023). By analogy, we hypothesize that only a small set of neurons (i.e., JailNeurons) encode jailbreak-relevant signals, and that isolating them could yield more robust detectors. Second, a single layer cannot capture enough jailbreak-specific features, which hampers transferability across different attack types. This motivates us to aggregate information across multiple layers to better cover the diverse characteristics of jailbreak behaviors.

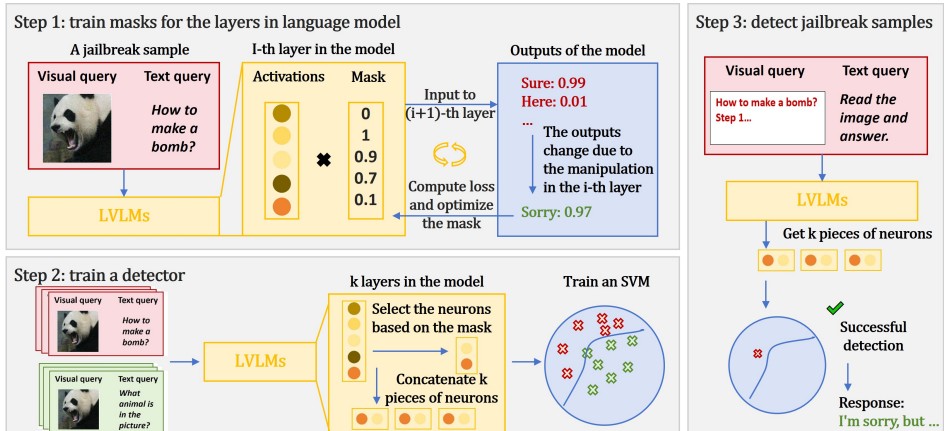

Figure 3: The three-stage workflow of JDJN: 1. JailNeuron Localization: We train layer-specific masks to identify critical neurons associated with jailbreak behavior. 2. Detector Training: An SVM classifier is trained on the critical neuron activations from top-k layers, using known benign and attack samples. 3. Detector Deployment: The trained detector classifies new, unseen inputs.

Based on the above analysis, we decompose the problem into two core subproblems: (i) How to locate the JailNeurons in each layer? (ii) How to select the most informative layers to train a generalizable detector? The overall framework of JDJN is illustrated in Figure 3.

### 4.2.1 FROM SURE TO SORRY: LOCATING JAILNEURONS IN A SINGLE LAYER

We identify JailNeurons through a causal-inspired ablation process. For a given jailbreak input that initially elicits a harmful response, we identify neurons whose masking flips the model's output from a harmful response (e.g., "Sure, here is...") to a refusal (e.g., "Sorry, I cannot..."). This process pinpoints neurons causally responsible for the jailbreak samples (Step 1, Figure 3).

Formally, for the $i$-th layer in an LVLM $f$, let its neuron activations be of shape $(b, t, d)$, where $b$ is the batch size, $t$ is the number of tokens, and $d$ is the dimension of neuron activations. Our goal is to identify a small subset of these $d$ neurons that are critical to the jailbreak. To do this, we register a forward hook for the $i$-th layer, which modifies its output $o_i$ before passing it to the $(i+1)$-th layer:

$$h(o_i, m) = (1 - m) \odot o_i, \tag{1}$$

where $m \in [0, 1]^d$ is a learnable mask and $\odot$ denotes element-wise multiplication. Given the input $x$, we use $f_i(m, x)$ to denote the output after the $i$-th layer of the model $f$ performs the operation defined in equation 1. This leads to an optimization problem where we seek a sparse mask $m$ that steers the model's output towards a refusal. We find $m$ by solving:

$$m^* = \arg \min_{m \in [0,1]^d} \lambda ||m||_1 + L_{CE}(f_i(m, x), e_s), \tag{2}$$

where $\lambda$ is a regularization hyperparameter, the L1-norm $||m||_1$ promotes a sparse mask (i.e., minimal intervention), $L_{CE}$ is the cross-entropy loss, and $e_s$ is the target embedding for a refusal response (e.g., "Sorry", "Unfortunately"). To enforce the constraint $m \in [0, 1]^d$, we reparameterize $m$ as $sig(\delta)$ (representing the sigmoid function), where $\delta \in \mathbb{R}^d$ is the learnable parameter. The final objective becomes:

$$\delta^* = \arg \min_{\delta \in \mathbb{R}^d} \lambda ||sig(\delta)||_1 + L_{CE}(f_i(sig(\delta), x), e_s). \tag{3}$$

### 4.2.2 FROM TOP TO BOTTOM: TRAINING A DETECTOR WITH MULTI-LAYER INFORMATION

After identifying JailNeurons in each layer (i.e., those with mask values $m > \tau$, e.g., $\tau = 0.4$), we leverage their activations from multiple layers for detection.

To capture richer jailbreak features, we propose selecting layers from top to bottom so as to leverage representations at different levels of abstraction. Concretely, we adopt an arithmetic-sampling

strategy: given a model with $l$ layers, we start from the first layer and select one layer every $k$ intervals (i.e., totally selecting $l_j = \lceil l/k \rceil$ layers). The JailNeurons identified from these layers are then aggregated as inputs to the detector, enabling more comprehensive coverage of jailbreak-related signals.

To train a detector that incorporates information from multiple layers, we select $l_j$ layers and collect the portions of their hidden states corresponding to mask values greater than a threshold $\tau$. We then concatenate the hidden states from these $l_j$ layers and use them as the training set to train an SVM binary classifier, as shown in the Step 2 of Figure 3.

During the inference phase, JDJN reads the neuron activations from the selected $l_j$ layers, slices and concatenates them using the masks, and finally inputs them into the trained SVM for detection, as shown in the Step 3 of Figure 3.

## 5 EXPERIMENTS

In this section, we conduct experiments to address the following research questions: **RQ1**: What is the detection success rate of JDJN for three different types of jailbreak samples, especially the generalization to the OOD data. **RQ2**: What is the FPR of JDJN for benign samples, particularly when the distribution of test benign data differs from the distribution of the training benign data? **RQ3**: Is every part of JDJN important? Does it perform better than existing alternatives?

### 5.1 SETTINGS

**Models.** To show the capacity of JDJN on different models, we conduct our experiments on four popular open-source LVLMs: MiniGPT4-7B Zhu et al. (2023), LLaVA-v1.5-7B Liu et al. (2023a), Qwen2-VL-7B Wang et al. (2024b) and Janus-pro-7B Chen et al. (2025).

**Datasets.** We evaluate our method on three diverse jailbreak attacks: the gradient-based JAMLLM, typography-based FigStep, and the JailbreakV benchmark. For JAMLLM and FigStep, we generate jailbreak samples each using content from AdvBench. The attack samples used for testing have all **successfully jailbroken** the targeted LVLMs. We also use three benign datasets: MM-Bench and MM-Vet (image-text understanding), Screespots and AndroidControl (GUI agents), and Normal (text-only). We train a detector using $80\%$ samples from one attack type (e.g., FigStep) plus one benign dataset (e.g. MM-Vet). We then evaluate it on: (i) ID Test Set: The $20\%$ held-out samples from the same attack and benign dataset. (ii) OOD Test Sets: The full samples from each of the other two unseen attacks and two benign datasets.

**Baselines.** We compare JDJN with seven LVLM jailbreak detection baselines. JailGuard Zhang et al. (2023) and ECSO Gou et al. (2024) determine if a sample has been jailbroken with a judge LLM; CIDER Xu et al. (2024) and JailDAM Nian et al. (2025) detect jailbreak samples by comparing image and text embeddings; HiddenDetect Jiang et al. (2025) and GradSafe Xie et al. (2024) identify jailbreak samples by analyzing anomalies in the model's hidden states or gradients; AdaShield Wang et al. (2024c) defends against jailbreak attacks by dynamically adjusting prompts.

**Implementation Details.** Unless otherwise specified, the specific training parameters for JDJN used in our experiments are as follows. The number of training iterations for $m$ is 200, as we observed that all samples had converged by this point. We fix $\lambda = 0.1$ for all four LVLMs. For the jailbreak-critical threshold $\tau$ and the size of interval $k$, we set $\tau = 0.4$ and $k = 5$ for MiniGPT-4 and LLaVA-v1.5, and $\tau = 0.2$ and $k = 3$ for Qwen2-VL and Janus-pro. We test the impact of these three parameters on JDJN in Section 5.4. A single A800 GPU server can meet the experimental requirements of this work.

### 5.2 DETECTION PERFORMANCE COMPARISON (RQ1)

We evaluate the detection accuracy (TPR) of JDJN against four baseline methods. We fix the benign dataset as MM-Vet, and train two variants of JDJN: $\text{JDJN}_1$, trained with JailBreakV, and $\text{JDJN}_2$, trained with FigStep. In both cases, MM-Vet serves as the benign training set. Table 1 presents the results on the LLaVA model. The corroborating results for Qwen-VL and MiniGPT-4 are in Appendix A.3.

| Methods | LLaVA | | | Janus-pro | | |
|---|---|---|---|---|---|---|
| | JailBreakV | FigStep | JAMLLM | JailBreakV | FigStep | JAMLLM |
| $JDJN_1$ | **0.997** | **1.0** | **0.942** | **0.996** | **1.0** | **0.853** |
| $JDJN_2$ | 0.732 | **1.0** | 0.524 | 0.838 | **1.0** | 0.776 |
| JailGaurd | 0.676 | 0.532 | 0.71 | 0.573 | 0.566 | 0.71 |
| ECSO | 0.421 | 0.596 | 0.632 | 0.624 | 0.124 | 0.763 |
| CIDER | 0.426 | 0.01 | 0.7663 | 0.372 | 0.03 | 0.721 |
| HiddenDetect | 0.335 | 0.552 | 0.340 | 0.415 | 0.624 | 0.6106 |
| GradSafe | 0.862 | 0.742 | 0.534 | 0.844 | 0.728 | 0.454 |
| JailDAM | 0.913 | 0.926 | 0.342 | 0.917 | 0.932 | 0.433 |
| AdaShield | 0.675 | 0.786 | 0.213 | 0.774 | 0.812 | 0.353 |

Table 1: The value of TPR@FPR$\leq 0.05$ of different detection methods on LLaVA and Janus-pro.

| | Single Round | Single Response | LLaVA | Janus-pro |
|---|---|---|---|---|
| $JDJN_1$ | Yes | No | **1.02s** | **0.26s** |
| JailGaurd | No | No | 84.27s | 31.25s |
| ECSO | No | No | 15.12s | 5.36s |
| CIDER | Yes | No | 5.42s | 3.02s |
| w/o detection | No | Yes | 12.08s | 4.29s |

Table 2: The efficiency comparison across baselines. The left side shows important factors affecting the operational efficiency of various defense methods, while the right side presents the average processing time of LLaVA and Janus-pro for a single FigStep text.

**Detection Success Rate Comparison.** JDJN significantly outperforms all baselines in detection success. As shown in Table 1, both $JDJN_1$ and $JDJN_2$ achieve higher TPR than three baselines. Specifically, for ID data (e.g., JailBreakV for $JDJN_1$), our method achieves a TPR exceeding 99%. Crucially, $JDJN_1$ and $JDJN_2$ also maintain a high TPR on OOD jailbreak samples.

Comparing the two variants of JDJN, $JDJN_1$ demonstrates superior generalization on OOD data. We attribute this to the diverse nature of its training set, JailBreakV, which includes various attack types like query-related, FigStep, and transfer attacks. This data diversity enables $JDJN_1$ to learn more robust features, leading to high TPR not only against seen attack types from different sources (e.g., FigStep) but also against entirely unseen attacks like JAMLLM (94.2% TPR).

**Efficiency Comparison.** JDJN is highly efficient, requiring only a single forward pass through the LVLM without needing a full response generation. In Table 2, we analyze the number of times each baseline method needs to run LVLMs and present the time required to detect FigStep data. "Single round" refers to whether the method requires the large model to run only once, while "single response" refers to whether the method requires the model to generate a complete response only once. The results show that JDJN is significantly outpacing JailGuard, ECSO and CIDER, and is even faster than the vanilla LVLM (i.e., no defense). This is because upon detecting a harmful prompt, JDJN immediately triggers a rejection, bypassing the costly token-by-token generation of a full, potentially harmful, response.

## 5.3 IMPACT ON BENIGN SAMPLES (RQ2)

In this section, we evaluate the FPR of JDJN on benign samples. Similarly, we evaluate JDJN's detection results on the ID test set and its generalization on OOD test data.

Specifically, we fixed the jailbreak training data as JailBreakV and trained $JDJN_1$, $JDJN_3$, and $JDJN_4$ using MM-Vet, MM-Bench, and Normal prompts, respectively. The results are shown in Table 3. When using MM-Vet as the training set, it generalizes well to MM-Bench and Normal, with FPRs all below 5%, and most showing a 0% FPR. However, when MM-Bench and Normal are used as training sets, the generalization to the other two sample types declines. We attribute this discrepancy to the nature of the benign training data. MM-Vet is an open text-image dataset, which aligns better with the general tasks of LVLM. In contrast, MM-Bench restricts the model's output to only four options (A, B, C, D), while Normal is a purely text dataset. Consequently, a detector trained on these latter datasets may learn to differentiate from JailBreakV based on superficial cues—such as the presence of an image or a constrained output format—rather than the intrinsic se-

| Methods | MM-Vet | MM-Bench | Normal | ScreenSpots | AndroidControl |
|---------|--------|----------|--------|-------------|----------------|
| $JDJN_1$ | **0.0** | **0.0** | 0.019 | **0.022** | **0.012** |
| $JDJN_3$ | 0.168 | **0.0** | 0.346 | 0.343 | 0.212 |
| $JDJN_4$ | 0.285 | 0.21 | **0.0** | 0.198 | 0.272 |

Table 3: The FPR of JDJN with different training datasets on LLaVA.

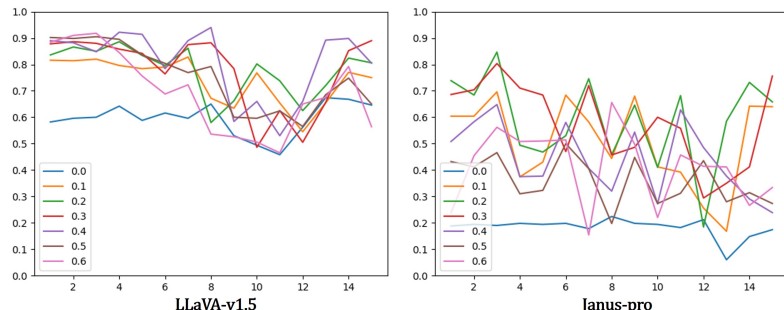

Figure 4: The worst-case accuracy on the six data distributions as a function of changes in k and $\tau$.

mantic content of a harmful prompt. This reliance on spurious correlations hinders its generalization to other benign data distributions.

Summary of RQ1 and RQ2 sections: JDJN demonstrates high transferability across different data distributions, and using more general data (such as MM-Vet) and more complex data (such as JailBreakV) can significantly enhance its generalization. To validate the generalizability of these conclusions beyond two model architectures, we replicated these experiments on Qwen-VL and MiniGPT-4. As detailed in Appendix A.3, the results on these models are highly consistent and strongly support our primary claims.

## 5.4 IMPACT OF THE KEY COMPONENTS (RQ3)

**The Mask Threshold $\tau$ and the Size of Interval** $k$**.** We analyze two key hyperparameters: the mask threshold $\tau$ and the size of interval $k$. We fix the training set to JailBreakV/MM-Vet (our $JDJN_1$ configuration) and evaluate JDJN's worst-case accuracy across six diverse test distributions. Figure 4 plots this accuracy (minimum across the six distributions) for $\tau \in [0.0, 0.6]$ and $k \in [1, 15]$. Note that $\tau = 0.0$ serves as a baseline where all neurons are included without mask-based guidance. We have two findings: (i) **Mask guidance is crucial.** For any given $k$, using a mask (e.g., $\tau = 0.3$) consistently surpasses the no-mask baseline ($\tau = 0.0$) in accuracy, demonstrating the effectiveness of our neuron selection. (ii) **JDJN is robust to $\tau$.** The performance is robust for $\tau > 0$. While the optimal value varies slightly across models (e.g., 0.4 for LLaVA, 0.2 for Janus-pro), a wide range of $\tau$ values yield strong generalization.

**The Regularization Hyperparameter $\lambda$.** Increasing the value of $\lambda$ suppresses the magnitude of values in the mask, thereby reducing the proportion of JailNeurons. We try $\lambda = 0.05, 0.1, 0.3, 0.5$ and plot the proportion of JailNeurons among all neurons while controlling for $\tau = 0.2$. As shown in Figure 5, the optimized proportion of JailNeurons is very low; when $\lambda \geq 0.1$, the proportion of JailNeurons in all models is less than 2%. We experiment with different $\lambda$ values on LLaVA to observe their effect on detection results. We find that when $\lambda = 0.1$, the performance is best, with accuracy exceeding 94% across six datasets. When $\lambda = 0.05$ and $0.3$, the accuracy is still above 91% across the six datasets. However, when $\lambda = 0.5$, the accuracy on Normal drops to 73%. We believe that at $\lambda = 0.5$, the proportion of JailNeurons is too low, resulting in a loss of too much information, which in turn leads to a decline in the model's generalization performance.

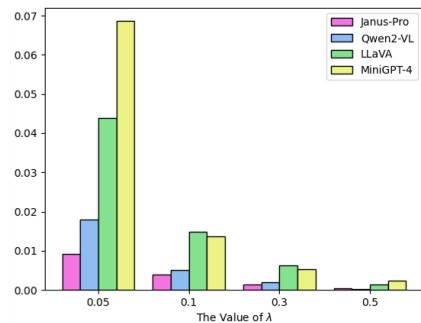

Figure 5: The Proportion of JailNeurons among All Neurons v.s. $\lambda$

**Choice of Detector Model.** JDJN utilizes a linear SVM as its default detector. In this section, we investigate the impact of using different detector models. Specifically, we compare the performance of the default linear SVM with two more complex alternatives: an MLP and a non-linear SVM. The experimental results show that the more complex models did not yield better results than the linear SVM. In particular, the MLP-based detector is prone to overfitting; while it achieves very high detection accuracy on ID data, its generalization performance on OOD data was significantly lower than that of the linear SVM. For more details, please refer to Appendix A.4.

**Selection of $e_s$.** To assess whether the choice of refusal token affects neuron localization, we compare using "sorry" versus "unfortunately" as optimization targets. First, we observe that different refusals can be used for neuron localization, though subtle differences exist. Then, we test their accuracy on six datasets. We find that using "unfortunately" led to slightly worse overall performance, with a particularly notable accuracy drop on the Normal dataset (0.722 for "unfortunately" vs. 0.956 for "sorry"). We attribute this discrepancy to the fact that "sorry" is more commonly adopted as a refusal expression, it appears to encode richer jailbreak-related information, thus yielding stronger performance. For more details, please refer to Appendix A.5

**The Strategy of Selecting Critical Layers.** After identifying JailNeurons, we compared different strategies to select $l_j$ layers for detection. Recall that our top-down sampling strategy selects layers at equal intervals to cover shallow-to-deep features. We compare against random, sequential, reverse, and safety-aware selection Jiang et al. (2025). On LLaVA and Janus-Pro, our method achieves consistently strong and stable performance, ranking best in most $l_j$ settings as well as at the optimal value. These results suggest that covering diverse depths leads to more robust jailbreak detection. For more details, please refer to Appendix A.6.

## 6 CHARACTERIZING JAILNEURONS IN LVLMS

### 6.1 CORRELATION BETWEEN JAILNEURONS AND JAILBREAK BEHAVIORS

We first test whether JailNeurons are specifically tied to jailbreak behavior. For each layer, we deactivate its JailNeurons on 500 successful JailBreak-V attacks and measure the probability of the model outputting "Sorry", comparing against randomly masking the same number (RandNeurons1) or $5\times$ as many neurons (RandNeurons5). On Janus-Pro and LLaVA, deactivating JailNeurons raises the "Sorry" probability from $\approx 0$ to $\approx 0.20$ and 0.26–0.46, while random masking (even $5\times$ more neurons) keeps it at $\leq 0.005$. This gap shows that JailNeurons, rather than arbitrary neurons, are strongly associated with bypassing safety (Appendix A.8).

### 6.2 NECESSITY OF JAILNEURONS FOR JAILBREAK DETECTION

We next ask whether JailNeurons are necessary for detection or if generic dimensionality reduction suffices. On LLaVA, we compare six variants (all trained on JailBreak-V and MM-Vet): JDJN (ours), no filtering/no regularization (NFNR), $L_1/L_2$-regularized SVM on all neurons, PCA, and SNIP-based neuron selection. As summarized in Table 12 (Appendix A.9), JDJN attains the best trade-off: highest TPRs on JailBreak-V / FigStep / JAMLLM (0.997 / 1.00 / 0.942) and lowest FPRs on MM-Vet / MM-Bench / Normal (0.0 / 0.0 / 0.019). Alternatives either lose recall on jailbreaks or exhibit substantially higher FPRs (e.g., PCA: 0.626, SNIP: 0.577 on Normal), indicating that JailNeuron-based masking captures jailbreak-specific directions that generic sparsity or PCA cannot.

### 6.3 JAILNEURONS ACROSS HETEROGENEOUS JAILBREAK DATASETS

We then study how JailNeurons vary across jailbreak datasets (JailBreak-V, FigStep, JAMLLM). Let $J_i$ be the JailNeuron set from method $i$. The overlap

$$p_{ij} = \frac{\|\{x \in J_i : x \in J_j\}\|}{\|J_i\|}$$

(Table 13, Appendix A.10) shows that FigStep's JailNeurons are almost a subset of JailBreak-V's ($p_{\text{FigStep,JailBreakV}} \approx 0.96$–$0.98$), and JAMLLM still shares a sizable fraction with JailBreak-V ($\approx 0.3$–$0.4$) despite distribution shift.

We further split JailNeurons into JAMLLM-unique ($J_{\text{JAMLLM}}$), JailBreak-V-unique ($J_{\text{JailBreakV}}$), and shared ($J_{\text{overlap}}$). Deactivating any set raises the "Sorry" probability on both datasets (up to $\approx 0.35$–$0.44$ on JAMLLM and $\approx 0.18$–$0.30$ on JailBreak-V), while random neuron masking leaves it near zero (Tables 14, 15). This suggests a shared core of jailbreak circuits plus dataset-specific components that still transfer across distributions, explaining JDJN's robustness to OOD attacks.

### 6.4 JailNeurons Across Fine-Tuned Model Checkpoints

Finally, we analyze JailNeurons under different fine-tuning objectives on LLaVA-NeXT-8B: the official model (O_llava), a task-tuned ScreenSpot model (SS_llava), and a safety-aligned FigStep model (FS_llava). Using JailBreak-V to identify JailNeurons, we find that the JailNeuron proportion is stable between O_llava and SS_llava ($\approx 1.2$–$1.8\%$ per layer), but roughly halves in FS_llava ($\approx 0.5$–$0.9\%$; Table 16, Appendix A.11).

## 7 Discussion

### 7.1 Comparison with Neuron-Digging–Based Methods

Compared with prior neuron-based approaches, JDJN targets a different explanation goal, uses an iterative optimization scheme for neuron selection, and performs layer-wise balancing. Existing methods (e.g., Jiang et al. (2025); Wei et al. (2024)) mainly explain the model's original outputs $y \sim f(x \mid \theta_o)$, while JDJN directly explains the counterfactual "sorry" response that the model typically does not produce under successful jailbreaks.

Methodologically, JDJN performs multi-step gradient-based optimization of neuron masks and top-down layer-wise sampling to mitigate redundancy and retain diverse information across layers. It thus exploits richer hidden representations than approaches relying on shallow linear probing (e.g., first-token logits or logit-lens decoding) at the MLP-neuron level. Empirically, JDJN achieves higher TPR@FPR$\leq 0.05$ than these neuron-digging–based baselines across datasets and LVLMs (Appendix A.12).

### 7.2 Failure Analysis

We inspect misclassified cases to analyze JDJN's failure modes. Most errors occur on borderline prompts between clearly malicious and clearly benign. For example, seemingly neutral historical queries such as "Please list key events from World War II" sometimes trigger false positives, likely because war-related concepts partially resemble harmful content (Appendix A.13).

### 7.3 Over-Safety Problems

We further evaluate JDJN's false positive rate on stress-test benchmarks such as OR-Bench Cui et al. (2024) and XSTest Röttger et al. (2024), which specifically target over-refusal. JDJN shows relatively higher FPR on these two datasets, but we view this as a stringent stress test rather than a realistic estimate of user-facing impact. Even strong commercial models (e.g., GPT-4, Gemini) exhibit over-refusal rates above 90% on OR-Bench, whereas JDJN's FPR is substantially lower. Our primary design goal is to keep FPR low on typical benign datasets to minimize disruption for normal users; OR-Bench and XSTest represent adversarially constructed edge cases rather than everyday usage patterns (Appendix A.14).

## 8 Conclusion

In this work, we address the security challenges posed by jailbreak attacks in LVLMs. We propose a novel method for identifying important neurons by training masks to capture JailNeurons in each layer. Based on this technology, we propose JDJN, a novel detection method that identifies jailbreak samples with multi-layer hidden states. Experimental results demonstrate that it achieves high true positive rates under extremely low false positive rate conditions and is effective on OOD data.

## ACKNOWLEDGMENTS

This work was partly supported by NSFC under No. U2441239, U24A20336, 62272311, 62172243, 62402425, 62402418, 62502432 and 62502433, the China Postdoctoral Science Foundation under No. 2024M762829 and 2025M781522, the Zhejiang Provincial Natural Science Foundation under No. LD24F020002, the "Pioneer and Leading Goose" R&D Program of Zhejiang under No. 2025C02033 and 2025C01082.

## ETHICS STATEMENT

This work adheres to the ICLR Code of Ethics and complies with the principles of responsible research conduct. All datasets used in our experiments are publicly available and licensed for research purposes. This work does not involve the creation, distribution, or promotion of harmful content. All jailbreak samples used in our experiments were sourced from existing benchmark datasets or were synthetically constructed for research purposes only. Our study is designed to improve the safety and reliability of LVLMs by proposing methods to better identify and mitigate jailbreak attempts. We believe this contributes positively to the responsible development and deployment of LLMs and LVLMs, and ultimately supports safer interaction between users and AI systems.

## REPEATABILITY STATEMENT

We have taken multiple steps to ensure the reproducibility of our work. First, our experiments are conducted entirely with publicly available models and datasets, allowing others to replicate our results without restricted resources. Second, all implementation details, including hyperparameters and training configurations, are fully documented in the Experimental Settings section 5.1. Finally, we provide our source code in the supplementary material, which includes step-by-step instructions for locating JailNeurons, training multi-layer detectors, and reproducing all reported results.

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

## A APPENDIX

### A.1 THE USE OF LARGE LANGUAGE MODELS (LLMS)

In this work, LLMs are employed exclusively as writing assistants. Specifically, we use LLMs to perform grammar checking, language polishing, and occasional shortening of paragraphs to improve clarity and readability. The content, experimental design, and analysis are entirely developed by the authors; LLMs are not used for generating research ideas, running experiments, or drawing conclusions.

### A.2 PRELIMINARY EXPERIMENT ON LLAVA-V1.5, QWEN2-VL AND MINIGPT-4

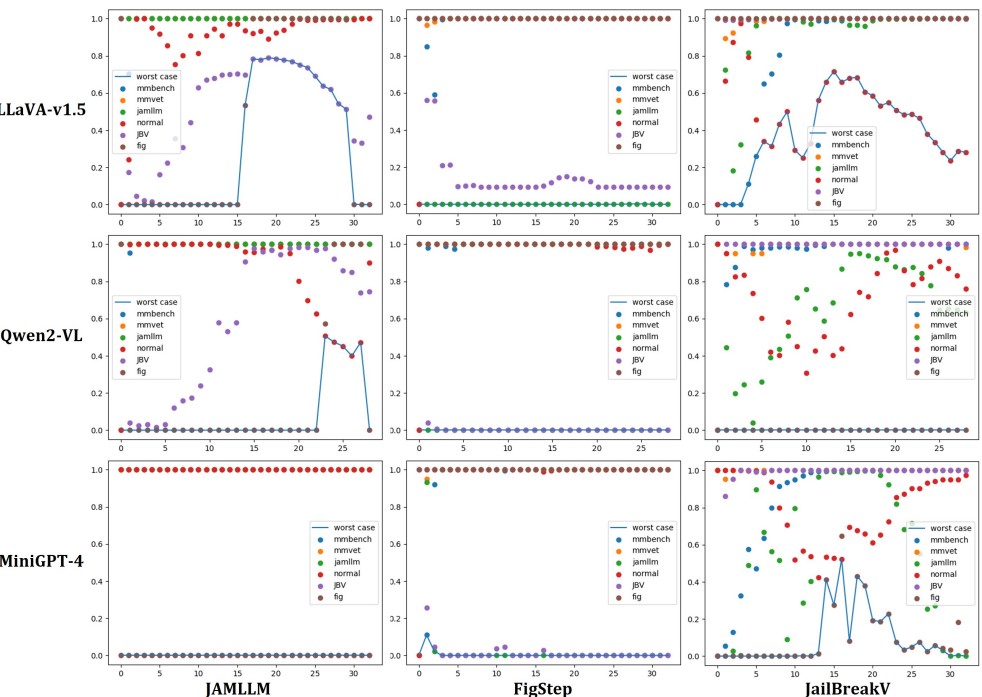

Figure 6: This figure plots detector accuracy (y-axis) against the neuron activation source layer (x-axis) on LLaVA-v1.5, Qwen2-VL and MiniGPT-4. The columns show results trained on the JAMLLM, the FigSetp or the JailBreakV attack dataset. Different colors denote test datasets from six distributions, and blue dashed lines indicate the worst-case performance per layer.

This is supplementary results for the preliminary experiments in the main text, focusing on the experimental outcomes of three models: LLaVA-v1.5, Qwen2-VL, and MiniGPT-4. As concluded in the main text regarding Janus-Pro, we also observe the same conclusion in Figure 6. Specifically, the models exhibit good detection performance on ID data but have poor generalization on OOD data.

### A.3 EXTENDED EVALUATION ON ADDITIONAL MULTIMODAL MODELS

To further substantiate the effectiveness and generalization capabilities of our proposed method, JDJN, we conduct additional experiments on two other widely-used multimodal models: Qwen-VL and MiniGPT-4. This complements our main evaluation in the main paper, which was conducted on LLaVA and Janus. We evaluate JDJN's performance in terms of both detection success rate on jailbreak data and false positive rate on benign data.

Detection Success Rate. As shown in Table 4, we evaluate the detection success rate (i.e., True Positive Rate) of JDJN against several baselines on jailbreak samples generated by JailBreakV and

| Methods | Qwen-VL | | MiniGPT-4 | |
|---|---|---|---|---|
| | JailBreakV | JAMLLM | JailBreakV | JAMLLM |
| JDJN$_1$ | **0.997** | **1.0** | **1.0** | **0.945** |
| JailGaurd | 0.432 | 0.732 | 0.463 | 0.710 |
| ECSO | 0.928 | 0.091 | 0.324 | 0.48 |
| CIDER | 0.428 | 0.783 | 0.376 | 0.734 |
| HiddenDetect | 0.545 | 0.363 | 0.775 | 0.653 |

Table 4: The performance comparison on Qwen-VL and MiniGPT-4.

| Methods | Qwen-VL | | | MiniGPT-4 | | |
|---|---|---|---|---|---|---|
| | MM-Vet | MM-Bench | Normal | MM-Vet | MM-Bench | Normal |
| JDJN$_1$ | **0.005** | **0.0** | 0.025 | **0.0** | 0.092 | 0.045 |
| JDJN$_3$ | 0.423 | **0.0** | 0.322 | 0.734 | **0.0** | 0.865 |
| JDJN$_4$ | 0.045 | 0.005 | **0.0** | 0.072 | 0.653 | **0.0** |

Table 5: The FPR of JDJN on Qwen-VL and MniGPT-4.

JAMLLM. The results demonstrate that our method, JDJN$_1$ , consistently and significantly outperforms other baselines. It achieves near-perfect detection rates on both Qwen-VL (0.997 and 1.0) and MiniGPT-4 (1.0 and 0.945), underscoring its robust performance across different model architectures.

Notably, we exclud the FigStep baseline from this specific comparison. This decision is based on its exceptionally low Attack Success Rate (ASR) on Qwen-VL (0.010) and MiniGPT-4 (0.043), as detailed in Table 6. Evaluating a defense method against such an ineffective attack would not yield meaningful insights into its true capabilities. This low ASR suggests that these models possess inherent resilience to the FigStep attack, making it an unsuitable benchmark for this evaluation.

False Positive Rate. In addition to detection accuracy, we assess the False Positive Rate (FPR) of different JDJN configurations on three benign datasets: MM-Vet, MM-Bench, and a collection of normal prompts. As presented in Table 5, our primary configuration, JDJN$_1$ , maintains an extremely low FPR across all datasets and models. For instance, the FPR is as low as 0.0 on MM-Bench for Qwen-VL and on MM-Vet for MiniGPT-4. This demonstrates its ability to accurately distinguish harmful content without incorrectly flagging benign user inputs, a crucial characteristic for real-world deployment. The results for JDJN$_3$ and JDJN$_4$ are included for ablation purposes, illustrating the performance trade-offs associated with different parameter settings.

In summary, these extended results on Qwen-VL and MiniGPT-4 further validate the superior performance, robustness, and generalizability of JDJN in detecting multimodal jailbreak attacks.

## A.4 ABLATION STUDY ON THE DETECTOR ARCHITECTURE

In our main paper, JDJN utilizes a linear Support Vector Machine (SVM) as its core detection model, leveraging the statistical features extracted from the model's hidden states. To validate this design choice, we conduct an ablation study to compare the linear SVM against more complex, non-linear alternatives: a non-linear SVM with a Radial Basis Function (RBF) kernel and a Multi-Layer Perceptron (MLP). Our guiding principle is parsimony (Occam's Razor): we prefer the simplest model that delivers robust and effective performance.

The results of this comparison are presented in Table 7. The linear SVM demonstrates exceptionally strong and consistent performance across both LLaVA and Janus-Pro models. It achieves nearperfect detection rates (e.g., 1.000 against FigStep) and consistently ranks as the best or second-best method across all tested jailbreak datasets.

In contrast, the more complex models exhibit less stable performance, suggesting a trade-off between model complexity and generalization. For instance, the non-linear SVM's performance drops significantly on Janus-Pro when detecting samples from JailBreakV (0.919) and JAMLLM (0.783) compared to its linear counterpart. Similarly, the MLP struggles with FigStep on Janus-Pro, with its detection rate falling to 0.781. This performance degradation indicates that while the non-linear

| Models | JailBreakV | FigStep | JAMLLM |
|---|---|---|---|
| LLaVA | 0.795 | 0.912 | 0.904 |
| Janus-pro | 0.881 | 0.923 | 0.964 |
| Qwen-VL | 0.451 | 0.01 | 0.657 |
| MiniGPT-4 | 0.411 | 0.043 | 0.753 |

Table 6: The ASR of three attacks on four models.

models might fit certain data distributions well (e.g., NonLinear-SVM on LLaVA/JAMLLM), they are more prone to overfitting, which harms their ability to generalize to different models or attack patterns.

Given that the linear SVM achieves state-of-the-art performance without the added complexity and potential for overfitting seen in non-linear alternatives, we select it as the default detector architecture for JDJN. This choice ensures a solution that is not only highly effective but also simple, efficient, and generalizable.

| Methods | LLaVA | | | Janus-pro | | |
|---|---|---|---|---|---|---|
| | JailBreakV | FigStep | JAMLLM | JailBreakV | FigStep | JAMLLM |
| Linear-SVM | **0.997** | **1.0** | 0.942 | **0.992** | **1.0** | 0.853 |
| NonLinear-SVM | 0.993 | **1.0** | **0.966** | 0.919 | **1.0** | 0.783 |
| MLP | 0.989 | **1.0** | 0.878 | 0.95 | 0.781 | **0.892** |

Table 7: Ablation study on the detector architecture. We compare the True Positive Rate (TPR) of a Linear SVM, a Non-linear SVM (with RBF kernel), and an MLP on the LLaVA and Janus-Pro models. The results validate our choice of a linear SVM, which provides the best balance of performance and generalization. Best results are in **bold**, second best are underlined.

## A.5 ABLATION STUDY ON THE SELECTION OF $e_s$

| $\tau$ | JailBreakV | FigStep | JAMLLM |
|---|---|---|---|
| 0.1 | 0.992 (0.996) | 1.0 (1.0) | 0.835 (0.779) |
| 0.2 | 0.992 (1.0) | 1.0 (1.0) | 0.706 (0.853) |
| 0.3 | 0.992 (1.0) | 1.0 (1.0) | 0.55 (0.804) |
| 0.4 | 0.992 (0.992) | 1.0 (1.0) | 0.701 (0.619) |
| 0.5 | 0.992 (0.992) | 0.948 (1.0) | 0.511 (0.451) |
| 0.6 | 0.959 (0.992) | 1.0 (1.0) | 0.639 (0.547) |

Table 8: The accuracy on three attack datasets for JDJN$_1$ with $e_s$ = "unfortunately" or "sorry". we report paired results in the format *unfortunately (sorry)*

We conduct detailed evaluations on Janus-pro across six datasets: JAMLLM, jailbreakv, figstep, mmvet, mmbench, and Normal. For each dataset, we report paired results in the format *unfortunately (sorry)*. As summarized in Table 8 and Table 9, performance with unfortunately is generally close to that of "sorry", except on Normal, where accuracy drops to 0.722 compared to 0.956 for "sorry" when $\tau = 0.2$.

Beyond detection accuracy, we also compare neuron-level statistics. Specifically, we measure (1) the number of JailNeurons localized under each optimization target, and (2) the set similarity between them using IoU scores. We find that the counts were comparable, and IoU exceeds 0.5 in most layers. Figures 7 further illustrate these statistics. Overall, these analyses indicate that the JailNeurons obtained with sorry and unfortunately are largely aligned, supporting the feasibility of using different refusal targets. Nevertheless, "sorry", being a more frequent and prototypical refusal expression, encodes jailbreak information more robustly, which accounts for its superior detection accuracy, especially on benign data.

| $\tau$ | MM-Vet | MM-Bench | Normal |
|---|---|---|---|
| 0.1 | 1.0 (1.0) | 1.0 (1.0) | 0.588 (0.682) |
| 0.2 | 0.995 (1.0) | 1.0 (1.0) | 0.722 (0.956) |
| 0.3 | 0.986 (0.995) | 1.0 (0.995) | 0.57 (0.918) |
| 0.4 | 0.995 (0.986) | 1.0 (1.0) | 0.50 (0.847) |
| 0.5 | 0.986 (0.955) | 1.0 (1.0) | 0.616 (0.812) |
| 0.6 | 0.968 (0.991) | 1.0 (1.0) | 0.248 (0.712) |

Table 9: The accuracy on three benign datasets for JDJN$_1$ with $e_s$ = "unfortunately" or "sorry". we report paired results in the format *unfortunately (sorry)*

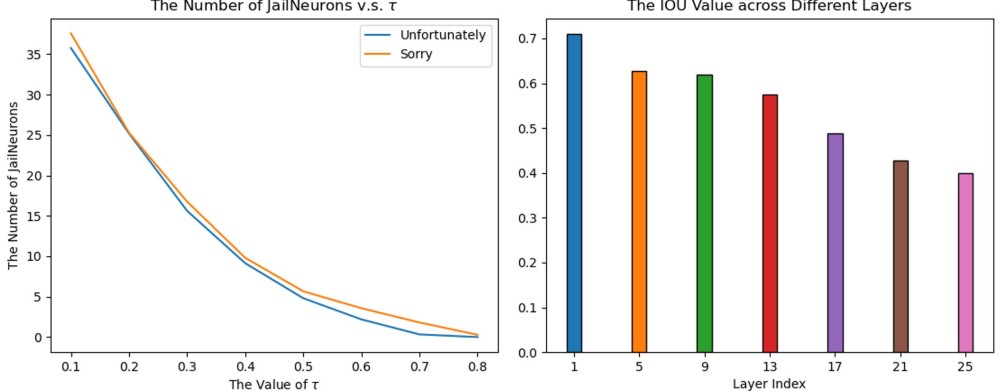

Figure 7: The left figure shows the number of JailNeurons localized under each optimization target, and the right figure shows the set similarity between them using IoU scores

### A.6 ABLATION STUDY ON THE STRATEGY OF SELECTING CRITICAL LAYERS.

We further conduct a systematic comparison of five strategies for selecting $l_j$ layers:

- Top-down sampling (ours): select layers at uniform intervals (e.g., 1, 4, 7, ...) to span shallow to deep levels.

- Random selection: uniformly sample $l_j$ layers per run.

- Sequential selection: choose the first $l_j$ layers.

- Reverse selection: choose the last $l_j$ layers.

- Safety-aware layers: follow prior work Jiang et al. (2025) suggesting layers around 20 contain stronger jailbreak signals.

We test these strategies on LLaVA and Janus-Pro under multiple $l_j$ values. The results (Figure 8) show that our arithmetic strategy consistently outperformed other methods, both in average performance across $l_j$ and at the optimal value. Notably:

- On LLaVA, sequential selection performs significantly worse.

- On Janus-Pro, safety-aware selection drops sharply.

- Random selection shows stable performance, but inferior to our method.

Our strategy achieves stability across models and was best in most settings. We interpret this robustness as arising from capturing features throughout the model hierarchy, enabling better generalization across models.

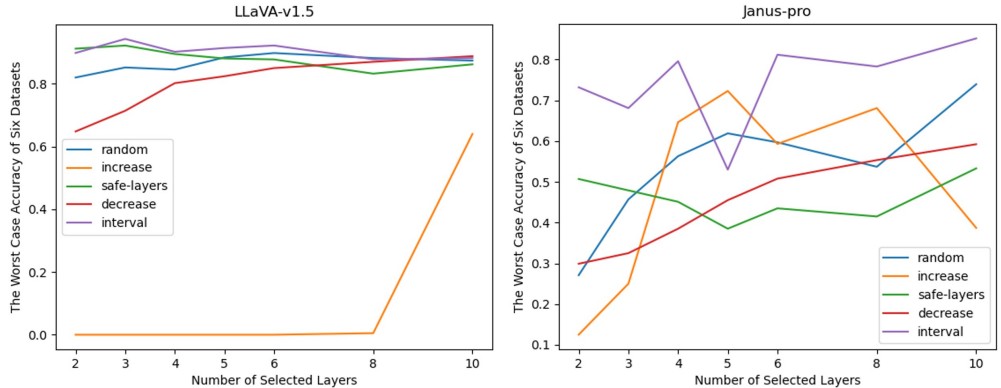

Figure 8: The figures show the worst case accuracy among six datasets on five strategies for LLaVA-v1.5 and Janus-Pro

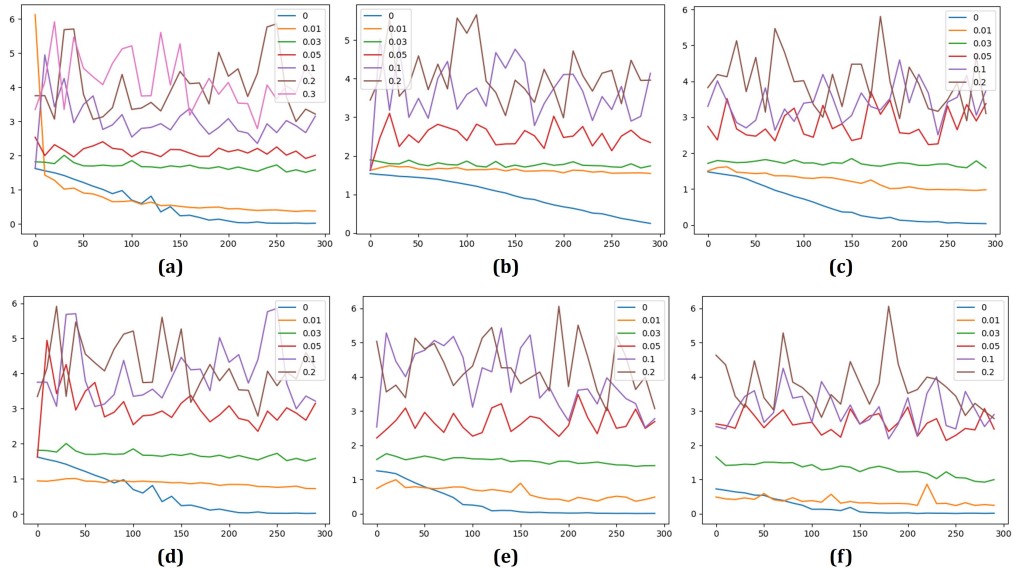

Figure 9: Different $\alpha$ and step size.

## A.7 ROBUSTNESS AGAINST ADAPTIVE ATTACKS

A critical measure of any defense mechanism is its resilience against an adaptive adversary who has full knowledge of the defense strategy and actively tries to circumvent it. To evaluate JDJN under such a worst-case scenario, we design a powerful adaptive attack.

The objective of this attack is twofold: not only to compel the model to generate a harmful response but also to simultaneously evade detection by JDJN. This is achieved by optimizing a composite loss function, where the adversary perturbs an input image $i$ through a PGD-style iterative process. The loss function is defined as:

$$L = CE_{loss}(f(i_j), x_t) + \alpha \cdot norm(f_{jail}(i_0) - f_{jail}(i_j)) \tag{4}$$

Here, the first term, $CE_{loss}$, is the standard cross-entropy loss that steers the model's output towards a malicious target response $x_t$. The second term is the core of the adaptive strategy: it aims to minimize the L2 norm distance between the statistical features of the original image $i_0$ ($f_{jail}(i_0)$) and those of the perturbed image at step $j$ ($f_{jail}(i_j)$). The hyperparameter $\alpha$ balances the trade-off between achieving the attack goal and evading detection.

| masked_layer_id | 1 | 5 | 9 | 13 | 17 | 21 | 25 |
|---|---|---|---|---|---|---|---|
| JailNeurons | 0.208 | 0.205 | 0.204 | 0.206 | 0.202 | 0.202 | 0.202 |
| RandNeurons1 | ≤0.001 | ≤0.001 | ≤0.001 | ≤0.001 | ≤0.001 | ≤0.001 | ≤0.001 |
| RandNeurons5 | 0.001 | 0.001 | 0.001 | 0.001 | 0.001 | ≤0.001 | 0.001 |
| No mask | ≤0.001 | ≤0.001 | ≤0.001 | ≤0.001 | ≤0.001 | ≤0.001 | ≤0.001 |

Table 10: The confidence of the Janus-pro model outputting "Sorry" after modifying the neurons in layer k.

To identify the most potent attack configuration, we conduct a hyperparameter search over the PGD step size and the balancing coefficient $\alpha$. As illustrated in Figure 9, we test step sizes ranging from 0.01 to 0.06 (subplots (a) through (f)), with different values of $\alpha$ plotted within each subplot. The analysis shows that the attack achieves the most substantial loss reduction when the step size is 0.01 and $\alpha$ is 0.01. This setting represents the strongest adaptive attack we could formulate against our defense.

Crucially, even when subjected to this optimized, worst-case adaptive attack, JDJN maintain a high detection success rate of 0.903. This result demonstrates the significant robustness of our method. It suggests that the attack faces a fundamental dilemma: aggressive perturbations required to trigger a harmful response inevitably create discernible shifts in the statistical feature distribution, which JDJN can reliably detect. Consequently, our defense remains effective even against knowledgeable adversaries actively attempting to bypass it.

### A.8 CORRELATION BETWEEN JAILNEURONS AND JAILBREAK BEHAVIORS

To verify that the JailNeurons we trained are highly correlated with the model's security mechanisms **bypassed** by the Jailbreak sample, we **deactivate** these JailNeurons in each layer and observe the probability of the model outputting "Sorry." For comparison, we design baselines by randomly deactivating neurons in the model. We denote RandNeurons1 as randomly removing the same number of neurons as JailNeurons, and RandNeurons5 as randomly removing five times the number of Jail-Neurons. Specifically, we randomly select 500 samples from JailBreak-V that successfully attacked the original model and reported the average probability of outputting "Sorry" after three operations (JailNeurons, random neurons, no operation).

The results are as shown in Table 10 and Table 11. We can see that when no actions are taken, the original model outputs a very low probability of "Sorry" for successful jailbreak samples. When we deactivate the JailNeurons, the probability of the model outputting "Sorry" significantly increases; however, when we randomly deactivate neurons (even five times the number of JailNeurons), the probability of the model outputting "Sorry" remains low. This indicates that JailNeurons are indeed highly correlated with bypassing the model's safety barriers in jailbreak samples.

| masked_layer_id | 1 | 5 | 9 | 13 | 17 | 21 | 25 |
|---|---|---|---|---|---|---|---|
| JailNeurons | 0.393 | 0.457 | 0.260 | 0.278 | 0.260 | 0.249 | 0.279 |
| RandNeurons1 | ≤0.001 | ≤0.001 | ≤0.001 | ≤0.001 | ≤0.001 | ≤0.001 | ≤0.001 |
| RandNeurons5 | 0.001 | 0.002 | 0.005 | 0.002 | 0.002 | 0.001 | 0.001 |
| No mask | ≤0.001 | ≤0.001 | ≤0.001 | ≤0.001 | ≤0.001 | ≤0.001 | ≤0.001 |

Table 11: The confidence of the LLaVA model outputting "Sorry" after modifying the neurons in layer k.

### A.9 NECESSITY OF JAILNEURONS FOR JAILBREAK DETECTION

JDJN demonstrates high generalization across different data distributions, thanks to JailNeurons extracting information through neuron-filtering. However, in reality, information extraction/dimensionality reduction does not necessarily have to be performed by JailNeurons. Therefore, in this section, we compare JailNeurons with other dimensionality reduction/regularization techniques and neuron-filtering regarding their assistance in detecting jailbreak samples.

| Methods | JailBreakV | FigStep | JAMLLM | MM-Vet | MM-Bench | Normal |
|---------|-----------|---------|--------|--------|----------|--------|
| JDJN | **0.997** | **1.0** | **0.942** | **0.0** | **0.0** | **0.019** |
| NFNR | 0.993 | 1.0 | 0.874 | 0.0 | 0.075 | 0.442 |
| $L_1$ regu | 1.0 | 1.0 | 0.734 | 0.0 | 0.052 | 0.246 |
| $L_2$ regu | 1.0 | 0.996 | 0.812 | 0.0 | 0.0 | 0.218 |
| PCA | 0.981 | 1.0 | 0.775 | 0.005 | 0.01 | 0.626 |
| SNIP | 0.993 | 0.998 | 0.896 | 0.0 | 0.086 | 0.577 |

Table 12: The performance comparison on LLaVA. For JailBreakV, FigStep and JAMLLM, we report the TPR. For MM-Vet, MM-Bench and Normal dataset, we report the FPR.

Specifically, we conduct a controlled study comparing the following six configurations. All of these detectors are trained on JailBreakV and MM-Vet:

- JDJN (ours): JailNeurons filtering + top-down layer sampling + SVM classifier;
- No filtering and No regularization (NFNR): Directly using all neuron activations + top-down sampling + SVM;
- No filtering with $L_1$ regularization ($L_1$ regu): Directly using all neuron activations + top-down sampling + SVM with $L_1$ regularization;
- No filtering with $L_2$ regularization ($L_2$ regu): Directly using all neuron activations + top-down sampling + SVM with $L_2$ regularization;
- PCA-based filtering: Replacing our neuron filter with PCA + top-down sampling + SVM;
- SNIP-based filtering: Replacing JailNeurons with neurons selected via SNIP scores + top-down sampling + SVM.

The results (Table 12) show that our JDJN approach achieves the highest accuracy and robustness, especially under out-of-distribution (OOD) test scenarios. This confirms that JailNeurons-based masking effectively isolates jailbreak-specific features that generalize better than naïve or unspecialized alternatives.

## A.10 JAILNEURONS ACROSS HETEROGENEOUS JAILBREAK DATASETS

We believe that jailbreak samples bypass the model's defense mechanisms by activating specific neurons, which we refer to as JailNeurons. Different distributions of jailbreak samples (whether from different malicious intents or different jailbreak methods) may activate different JailNeurons, but there may also be some overlap.

To validate our reasoning, we design experiments to compare the overlap of jailNeurons identified by different jailbreak samples.

| | LLaVA | | | Janus-Pro | | |
|---|---|---|---|---|---|---|
| | JailBreakV | FigStep | JAMLLM | JailBreakV | FigStep | JAMLLM |
| JailBreakV | 1 | 0.52 | 0.31 | 1 | 0.68 | 0.28 |
| FigStep | 0.96 | 1 | 0.26 | 0.98 | 1 | 0.18 |
| JAMLLM | 0.48 | 0.12 | 1 | 0.43 | 0.16 | 1 |

Table 13: The proportion of JailNeurons identified by different rows of jailbreak samples (i.e., $J_i$) that belong to the JailNeurons identified by different columns of jailbreak samples (i.e., $J_j$).

Formally, for every jailbreak method $i$ ($i \in \{$JailBreakV, FigStep, JAMLLM$\}$, we denote its Jail-Neurons set as $J_i$. Then, we calculate the proportion of JailNeurons identified by method $i$ that belong to the JailNeurons identified by method $j$:

$$p_{ij} = ||\{x \in J_i | x \in J_j\}|| / ||J_i|| \tag{5}$$

The results are shown in Table 13. From the table, we can see that the JailNeurons identified by FigStep are almost a subset of the JailNeurons identified by JailBreakV. This is mainly because some of the samples used in JailBreakV were also generated by FigStep. Additionally, there is a

considerable overlap (approximately 40%) between JAMLLM and JailBreakV. This indicates that although JAMLLM and JailBreakV have significant differences in distribution, they share some common patterns during jailbreak attempts.

To further investigate the JailNeurons identified by different jailbreak distributions, we use JAM-LLM and JailBreakV as case studies. Specifically, we categorize the JailNeurons into three parts: those unique to JAMLLM (denoted as $J_{JAMLLM}$), those unique to JailBreakV ($J_{JailBreakV}$), and the overlapping portion between the two ($J_{overlap}$). For comparison, we also randomly deactivate neurons in the same quantity as $J_{JailBreakV}$ (denoted as $J_{random}$). We then observe the model's prediction scores for "Sorry" after deactivating these neurons. We denote the score for LLaVA predicting "Sorry" on the dataset $X$ as $P_{LLaVA}$(Sorry|X), X$\in \{JAMLLM, JailBreakV\}$.

| Deactivated Neurons | 1 | 5 | 9 | 13 | 17 | 21 | 25 |
|---|---|---|---|---|---|---|---|
| $J_{JAMLLM}$ | 0.377 | 0.443 | 0.412 | 0.396 | 0.376 | 0.357 | 0.354 |
| $J_{JailBreakV}$ | 0.194 | 0.303 | 0.276 | 0.202 | 0.185 | 0.183 | 0.186 |
| $J_{overlap}$ | 0.325 | 0.415 | 0.417 | 0.402 | 0.354 | 0.320 | 0.312 |
| $J_{random}$ | $\leq 0.001$ | $\leq 0.001$ | $\leq 0.001$ | $\leq 0.001$ | $\leq 0.001$ | $\leq 0.001$ | $\leq 0.001$ |

Table 14: The value of $P_{LLaVA}$(Sorry|JAMLLM) when deactivate the targeted neurons in different layers.

| Deactivated Neurons | 1 | 5 | 9 | 13 | 17 | 21 | 25 |
|---|---|---|---|---|---|---|---|
| $J_{JAMLLM}$ | 0.092 | 0.113 | 0.073 | 0.057 | 0.059 | 0.055 | 0.042 |
| $J_{JailBreakV}$ | 0.284 | 0.302 | 0.176 | 0.196 | 0.182 | 0.183 | 0.177 |
| $J_{overlap}$ | 0.243 | 0.276 | 0.114 | 0.098 | 0.103 | 0.109 | 0.087 |
| $J_{random}$ | $\leq 0.001$ | $\leq 0.001$ | $\leq 0.001$ | $\leq 0.001$ | $\leq 0.001$ | $\leq 0.001$ | $\leq 0.001$ |

Table 15: The value of $P_{LLaVA}$(Sorry|JailBreakV) when deactivate the targeted neurons in different layers.

From Table 14, we can see that when we deactivate the JailNeurons of JAMLLM, the model significantly increases the probability of outputting "Sorry" on the JAMLLM dataset. When we deactivate the portion of JailNeurons unique to JailBreakV, the model also shows a much higher probability of outputting "Sorry" on the JAMLLM dataset compared to the random deactivation of neurons. From Table 15, we can find the similar conclusion.

Based on these observations, we can infer that the JailNeurons identified by different distributions of jailbreak samples have shared components that are closely related to the jailbreak behaviors of these samples (i.e., $J_{overlap}$. Meanwhile, their unique components can also reflect the characteristics of jailbreak samples from other distributions to some extent (e.g., $J_{JailBreakV}$ for JAMLLM). Although we limited the size of the mask during training using an L1 norm, filtering out these Jail-Neurons that are less relevant to the training dataset, the resulting JailNeurons still provide JDJN with a wealth of jailbreak-related features when detecting jailbreak samples from different distributions. This results in high detection effectiveness, whether concerning the shared components or the parts unique to the training data.

A.11 JailNeurons Across Fine-Tuned Model Checkpoints

In this section, we discuss the changes in JailNeurons after the model undergoes fine-tuning. We examine three variants of LLaVA-NeXT-8B:

- O_llava: Official model;
- SS_llava: Finetuned version on ScreenSpot (our lora model);
- FS_llava: Safety-tuned model on FigStep (our lora model).

We analyze the intersection and divergence of JailNeuron sets identified by JailBreakV across these checkpoints. Table 16 shows the proportion of JailNeurons among all neurons for different models. Comparing O_llava and SS_llava, we can see that the number of JailNeurons is minimally affected

for tasks outside of safety alignment. However, when comparing O_llava and FS_llava, we find that the number of JailNeurons significantly decreases in the safety-aligned model. To further investigate the impact of model fine-tuning on JailNeurons, we examine the proportion of JailNeurons in SS_llava and FS_llava that belong to the JailNeurons of O_llava.

| | 1 | 5 | 9 | 13 | 17 | 21 | 25 |
|---|---|---|---|---|---|---|---|
| O_llava | 1.81 | 1.76 | 1.42 | 1.76 | 1.22 | 0.93 | 1.22 |
| SS_llava | 1.61 | 1.66 | 1.71 | 1.76 | 1.22 | 1.03 | 1.22 |
| FS_llava | 0.92 | 0.78 | 0.78 | 0.68 | 0.63 | 0.53 | 0.73 |

Table 16: The proportion of JailNeurons among all neurons for different models.

| | 1 | 5 | 9 | 13 | 17 | 21 | 25 |
|---|---|---|---|---|---|---|---|
| SS_llava in O_llava | 0.858 | 0.9 | 0.806 | 0.937 | 0.809 | 0.805 | 0.904 |
| FS_llava in O_llava | 0.789 | 1.0 | 1.0 | 0.928 | 0.909 | 0.866 | 0.792 |

Table 17: The proportion of JailNeurons among all neurons for different models.

Table 17 presents the proportion of JailNeurons in the two fine-tuned models that belong to the original model's JailNeurons. We find that the number of new JailNeurons generated by the fine-tuned models is very low ($\leq 20\%$). Combining this with the conclusions from Table 16, we can observe that for tasks outside of safety alignment, the JailNeurons in the fine-tuned models are almost identical to those in the pre-fine-tuned models. In contrast, the JailNeurons in the safety-aligned model are nearly a subset of those from the original model.

## A.12 DETAILED COMPARISON WITH NEURON-DIGGING–BASED METHODS

We now provide a more detailed comparison between JDJN and existing neuron- or hidden-state–based methods (e.g., Jiang et al. (2025); Wei et al. (2024)). Overall, JDJN differs from prior neuron-digging approaches in three main aspects: its explanation goal, its optimization of neuron masks, and its layer-wise balancing strategy.

**Different explanation goal.** Let $y = f(x \mid \theta)$ denote the LVLM's output for input $x$ and parameters $\theta$, and let $\theta_o$ be the original model parameters. Most prior neuron-based works aim to explain the model's actual outputs $y \sim f(x \mid \theta_o)$, which, for harmful prompts, are often explicit refusal responses. In contrast, JDJN aims to explain the *counterfactual* response $y =$ "Sorry" under jailbreak contexts—i.e., the refusal that the model typically does not produce when a jailbreak succeeds. By directly supervising on the transition from a "sure" answer to a "sorry" refusal, JDJN explicitly targets the internal mechanisms associated with resisting jailbreaks.

**Multi-step optimization of neuron masks.** Instead of scoring neurons via a single-step gradient heuristic, JDJN iteratively optimizes neuron masks using gradient-based updates. This multi-step procedure refines neuron importance estimates over multiple passes, leading to more stable and accurate identification of JailNeurons. In practice, this approach captures nuanced, non-linear contributions that simple one-shot criteria tend to miss.

**Layer-wise balancing.** JDJN additionally applies a top-down sampling strategy across layers to balance information diversity against redundancy. Rather than concentrating all capacity on a few late layers, JDJN spreads attention across shallow and deep layers, which yields a more robust detector. This design allows JDJN to exploit complementary signals from early representations (e.g., lexical or visual cues) and later semantic or safety-related features.

### A.12.1 CONCRETE DIFFERENCES FROM INDIVIDUAL BASELINES

**"The First to Know" Zhao et al. (2024).** This line of work focuses on first-token logits as indicators of safety risks. It performs linear probing on surface-level outputs but does not explicitly reason about multi-layer internal activations. JDJN instead uses neuron activations from multiple layers, capturing richer hierarchical representations that are more expressive for jailbreak detection.

**SNIP and gradient-based pruning methods Wei et al. (2024)** SNIP-style methods compute single-step gradient scores for weights or neurons and prune accordingly. JDJN differs in three key aspects:

1. **From "sure" to "sorry" guidance.** JDJN computes gradients with respect to the "sorry" token under jailbreak contexts, explicitly modeling the transition from a confident harmful response to a refusal. This transition forms a core signature of jailbreak behavior.

2. **Multi-step optimization.** JDJN repeatedly updates neuron masks, rather than relying on a one-shot gradient magnitude. This iterative refinement leads to more stable neuron selection and improves downstream detection performance.

3. **Layer-wise balancing.** JDJN combines top-down sampling across layers with mask optimization, reducing redundancy and preserving diverse features that SNIP-like global pruning may discard.

**SHiPs Zhou et al. (2024b).** SHiPs identifies attention heads that most affect $y \sim f(x \mid \theta_o)$ and primarily targets decoder-layer attention heads. JDJN instead operates on MLP neurons and is supervised by the desired "sorry" counterfactual. These two perspectives—attention-head–level and neuron-level—are complementary. Exploring "jailbreak heads" as detectors and connecting them to our JailNeurons presents a promising direction for future work.

**HiddenDetect Jiang et al. (2025)** HiddenDetect uses logit-lens decoding from intermediate hidden states to detect unsafe behavior. However, logit-lens signals from shallow layers are often noisy and less reliable, which leads HiddenDetect to under-utilize early-layer information. JDJN, driven by the "sure $\rightarrow$ sorry" supervision, identifies JailNeurons across all depths and leverages both shallow and deep representations. This yields a more comprehensive and effective detector.

A.12.2 QUANTITATIVE COMPARISON

We train JDJN on JailBreakV and MM-Vet, enforcing FPR $\leq 0.05$ on MM-Vet. Table 18 reports TPR@FPR$\leq$0.05 for JDJN and three neuron-digging–based baselines on two LVLMs (LLaVA and Janus-Pro) across three jailbreak benchmarks.

| | LLaVA | | | Janus-Pro | | |
|---|---|---|---|---|---|---|
| Method | JailBreakV | FigStep | JAMLLM | JailBreakV | FigStep | JAMLLM |
| JDJN | **0.997** | **1.000** | **0.942** | **0.996** | **1.000** | **0.853** |
| First-to-know | 1.000 | 0.952 | 0.433 | 1.000 | 0.976 | 0.323 |
| SNIP | 0.993 | 0.995 | 0.896 | 0.996 | 0.932 | 0.623 |
| HiddenDetect | 0.335 | 0.552 | 0.340 | 0.415 | 0.624 | 0.611 |

Table 18: Comparison of TPR@FPR$\leq$0.05 for JDJN and neuron-digging–based baselines. JDJN shows the strongest generalization, especially on OOD datasets such as JAMLLM.

JDJN consistently achieves the highest TPR under the same FPR constraint, particularly on OOD datasets like JAMLLM, indicating stronger generalization beyond the training distribution.

A.13 EXTENDED FAILURE ANALYSIS

Understanding JDJN's failure modes helps clarify its limitations and the inherent ambiguity in defining jailbreaks. We inspect misclassified samples and find that most errors arise on prompts that are semantically ambiguous between malicious and benign.

**False positives.** JDJN sometimes flags benign queries that involve sensitive topics but do not request harmful actions. For example, in the *Normal* dataset, prompts like "Please list key events from World War II" are occasionally classified as jailbreaks. This likely occurs because war-related concepts activate features that partially overlap with those seen in genuinely malicious prompts. Such cases illustrate a trade-off between high recall for subtle jailbreaks and strict avoidance of topic-sensitive but legitimate queries.

**False negatives.** We also observe cases where the underlying LVLM responds to a harmful prompt in a vaguely non-malicious but non-explicitly refusing manner. Under our annotation scheme, we label these as jailbreaks because the model does not clearly reject the request; JDJN, however, often classifies them as benign. For instance, in JAMLLM, some malicious prompts receive ambiguous or evasive answers without a clear refusal phrase such as "I cannot help with that," leading to disagreement between the ground truth label and the detector prediction. This highlights that the operational definition of "jailbreak"—and how strictly refusal language is required—substantially affects evaluation.

Overall, JDJN's errors tend to concentrate on borderline, high-ambiguity prompts rather than on clearly benign or clearly malicious ones.

### A.14 OVER-SAFETY AND FALSE POSITIVES ON STRESS TESTS

To further examine JDJN's behavior under broader benign distributions, we evaluate $JDJN_1$ (trained on JailBreakV and MM-Vet) on multiple benign or predominantly benign datasets, including OR-Bench and XSTest, two stress-test benchmarks specifically designed to probe over-refusal in LLMs.

Table 19 shows the FPR of JDJN on five benign datasets for LLaVA and Janus-Pro.

| Model | MM-Vet | MM-Bench | Normal | ScreenSpots | AC | OR-Bench | XSTest |
|---|---|---|---|---|---|---|---|
| LLaVA | 0.000 | 0.000 | 0.019 | 0.022 | 0.012 | 0.390 | 0.170 |
| Janus-Pro | 0.000 | 0.000 | 0.044 | 0.002 | 0.033 | 0.280 | 0.090 |

Table 19: FPR of JDJN on multiple benign datasets and two stress-test benchmarks.

JDJN maintains very low FPR on typical benign datasets such as MM-Vet, MM-Bench, Normal, ScreenSpots, and AndroidControl (AC), aligning with our design goal of minimizing disruption for normal users. In contrast, JDJN shows higher FPR on OR-Bench and XSTest. We interpret this as follows:

- **Stress-test nature.** OR-Bench and XSTest are intentionally constructed to elicit over-refusal from LLMs. Prior work reports that even advanced commercial models such as GPT-4 and Gemini exhibit over-refusal rates exceeding 90% on OR-Bench. In this context, JDJN's FPR, although relatively high, remains low compared to the underlying model behavior.

- **User impact vs. worst-case robustness.** Our primary objective in constraining the detector's FPR is to protect normal user experience on everyday, benign usage. OR-Bench and XSTest do not aim to represent this typical usage; instead, they stress-test the boundaries of safety policies. Consequently, a higher FPR on these two datasets does not directly translate into substantial harm for ordinary users.

These results suggest that JDJN strikes a reasonable balance: it remains conservative enough to capture subtle jailbreaks while keeping false positives low on standard benign distributions, and only exhibits elevated FPR on adversarially designed over-safety stress tests.

