# OpenReview forum: "From ``Sure" to ``Sorry": Detecting Jailbreak in Large Vision Language Model via JailNeurons"
_ICLR.cc/2026/Conference — ICLR 2026 Poster_

### Official Review · Reviewer_pMai · 2025-10-27

**Soundness:** 2
**Presentation:** 3
**Contribution:** 3
**Rating:** 4
**Confidence:** 4

**Summary:**

This paper presents a novel method to locate "jailneurons" in MLLMs so that the use of a classifier (in this paper, the authors adopt SVM) could handle the jailbreak detection. In detail, to finish the detection, they first locate related neurons by an optimization process and concatenate these neurons for SVM training. Experiments on several models, as well as a few jailbreak methods, demonstrate the performance of such a method.

**Strengths:**

- Clear writing. The pipeline, mathematical details, and experimental setup are clear, which is easy to follow.
-  Novel way to locate the neurons. The method to build an optimization framework for neuron selection is interesting.

**Weaknesses:**

- My biggest concern is about the experimental results. Take the evaluation on FigStep as an example. ECSO got 90.3% on the OCR subset of MM-safetybench (according to the original paper), which is a similar jailbreaking dataset using typography, so it seems a little weird that ECSO only got 0.596 on FigStep. Besides, the performance in the HiddenDetect paper is 0.846, where it also shows the performance of CIDER on Figstep is 0.713. The numbers reported in Table 1 have a huge gap with the original papers (or other replications), which requires further explanation.
- From the perspective of storytelling, it does not clearly state the difference between previous neuron-digging methods and this script. Such detection could indeed be facilitated via a simple classifier, which is more efficient than using a guardrail model, but it is the advantage of all similar methods, such as SNIP, HiddenDetect, etc. The focus should be on the disadvantages of the previous safety-neuron picking method, or their suboptimal layer-picking method. More comparisons on this line of work are required[1][2][3] to prove that previous works could only handle normal harmful requests, other than jailbreaking requests.
- The figures are not clear. More information should be included in the caption or related text parts. For example, what is the value in Table 1? I finish my review with the hypothesis that it is the detection rate (successfully detected/all jailbreak samples)

[1]The First to Know: How Token Distributions Reveal Hidden Knowledge in Large Vision-Language Models?

[2]Assessing the Brittleness of Safety Alignment via Pruning and Low-Rank Modifications.

[3] On the Role of Attention Heads in Large Language Model Safety

**Questions:**

- What is the detailed experimental setup of baselines?
- Could you explain more about the difference between previous neuron-digging methods and this script?
- Will this method be over-sensitive, i.e., classifying benign prompts as jailbreaking? Experiments on or-bench or XSTest (or other MLLM datasets, if any) would be better.

---

> ### Author Response · Authors · 2025-11-21
> **Author Response (part 1 of 3)**
>
> ### 1. Clarification on baseline reproduction and discrepancies with original papers
>
> **Our response.**
>
> We thank the reviewer for carefully checking the reported numbers and for pointing out these discrepancies. All baselines are implemented following their official papers and/or released codebases as closely as possible. The differences from the original reports mainly arise from three methodological choices in our evaluation protocol:
>
> 1.  **Evaluation metric.**
>
> We focus on the true positive rate under a low false positive rate, because we aim for detectors that do not noticeably degrade normal user experience. We report the TPR with FPR ≤ 0.05 as a common constraint [1]. This differs from settings where AUROC or TPR at a higher FPR are reported. As a result, our TPR numbers are generally lower but reflect a stricter operating point.
>
> 2.  **Selection of jailbreak samples.**
>
> In our experiments, we only evaluate on **successful jailbreaks**. We regard a detector as a complement to safety alignment, so we exclude prompts that the LVLM already rejects. Including such cases would inflate detection performance, since they often contain explicit refusal signals.
>
> 3.  **Criterion for successful jailbreak.**
>
> We define a jailbreak as successful if the model’s output does **not** any pre‑specified refusal strings (including "Sorry", "I'm sorry", "Unfortuantely", "As an assistant"...), whereas some prior work (e.g., ECSO) uses a judge model (such as GPT‑4) to decide whether a response is harmful. Both criteria have pros and cons; in this work, we follow the previous works [2] to choose a fully automatic criterion based on explicit refusal markers. This causes numerical differences compared with evaluations that rely on a human‑level judge.
>
> Below we clarify these effects for ECSO, CIDER, and HiddenStates on FigStep in more detail.
>
> -  **ECSO.**
>
> ECSO relies on the model’s self‑reflection: the same model is asked whether its generated output is harmful. Its performance thus depends on
>
> (i) how explicit harmful signals are in the output, and
>
> (ii) the model’s own ability to recognize them.
>
> In FigStep, malicious instructions are decomposed into multiple innocuous‑looking steps (e.g., “Generate descriptive contents for items 1–3…”), which substantially weakens the model’s self‑assessment capabilities and thus ECSO’s performance. In addition, the original ECSO paper uses GPT‑4 as a judge for harmfulness, whereas we evaluate success using refusal strings. This difference in success criteria leads to a different subset of “successful” jailbreaks and hence different reported scores.
>
> -  **CIDER.**
>
> CIDER detects attacks by comparing embedding similarities between text prompts and images before and after denoising, and is particularly sensitive to optimization‑based attacks. As also discussed in its original paper, it is less robust to structurally deceptive prompts such as FigStep.
>
> The discrepancy with numbers reported in the HiddenDetect paper is mainly due to **thresholding**: in our setting, we **fix the threshold to ensure FPR ≤ 0.05** on benign data, while the threshold selection procedure in the HiddenDetect paper is not fully specified. This stricter FPR constraint naturally leads to lower TPR.
>
> -  **HiddenDetect.**
>
> HiddenDetect uses intermediate‑layer semantics decoded via a logit lens, under the assumption that some layers still encode refusal signals even when the final answer is unsafe. Our results differ for two main reasons:
>
> 1. We evaluate **only successful jailbreak samples**, excluding failed ones which typically contain strong refusal signals and are easier to detect. This stricter filtering lowers the observed TPR.
>
> 2. HiddenDetect reports AUROC to characterize performance over all thresholds, whereas we report **TPR at the specific threshold where FPR ≤ 0.05** on MM‑Vet. Fixing a low‑FPR threshold leads to more conservative TPR.
>
> Overall, our setup emphasizes **low‑FPR, successful‑only jailbreak detection** controlled by a transparent refusal‑string criterion, which systematically yields different numbers across all methods.

---

> > ### Author Response · Authors · 2025-11-21
> > **Author Response (part 2 of 3)**
> >
> > ### 2. Difference between JDJN and existing neuron‑digging–based methods
> >
> >
> > **Our response.**
> >
> > We appreciate this suggestion and agree that a clearer comparison with prior neuron‑based approaches ([1]–[3] and HiddenDetect) helps sharpen our contribution. JDJN differs from existing neuron‑digging methods in three main aspects:
> >
> > 1.  **Different explanation target.**
> >
> > Let $y = f(x \mid \theta)$ be the output of an LVLM with parameters $\theta$ and input $x$. Most previous neuron‑based safety works aim to analyze outputs under the **original behavior** $y \sim f(x \mid \theta_o)$, where the model often produces refusal tokens for normal harmful prompts. In contrast, JDJN explicitly focuses on explaining the **counterfactual refusal behavior** $y = \text{“Sorry”}$ in **jailbreak contexts**, where the model typically does _not_ refuse. This “what the model should have said” objective leads to a different set of neurons (i.e., “JailNeurons”) compared with those identified for standard harmful prompts.
> >
> > 2.  **Multi‑step optimization of neuron masks.**
> >
> > JDJN uses an iterative gradient‑based optimization of binary neuron masks, rather than a single‑step importance scoring. This multi‑step optimization allows us to refine neuron importance scores over several iterations and to better isolate neurons that causally support the desired “sorry” behavior under jailbreak prompts.
> >
> > 3.  **Layer‑wise balancing.**
> >
> > JDJN introduces a top‑down sampling scheme to balance neuron selection across layers. This explicitly trades off information diversity against redundancy, ensuring that the final neuron set covers both shallow and deep representations, which empirically leads to stronger generalization of the detector.
> >
> > Below we discuss concrete differences with representative baselines referenced by the reviewer.
> >
> > -  **[3] “The First to Know”.**
> >
> > This method mainly focuses on first‑token logits as early indicators of potential safety risks and applies relatively shallow linear probing. JDJN instead aggregates **multi‑layer neuron activations** guided by the “sure → sorry” supervision signal, thereby capturing richer hierarchical representations and enabling more robust jailbreak detection.
> >
> > -  **[4] SNIP and related gradient‑based pruning methods.**
> >
> > While SNIP also uses gradient information for neuron, JDJN differs in several ways:
> >
> > 1.  **Gradient objective.** SNIP evaluates the effect of pruning on preserving the original output, whereas JDJN computes gradients with respect to the **refusal token (“sorry”) under jailbreak prompts**, explicitly modeling the transition from unsafe “sure” behavior to safe “sorry” behavior.
> >
> > 2.  **Optimization procedure.** SNIP relies on a single forward–backward pass to score parameters, while JDJN performs **multi‑step optimization of neuron masks**, leading to more stable neuron selection.
> >
> > 3.  **Layer allocation.** JDJN additionally incorporates layer‑wise balancing to prevent over‑concentration on a few layers, which we find important for cross‑dataset generalization.
> >
> > -  **[5] SHiPs (ICLR ’25).**
> >
> > SHiPs focuses on identifying **attention heads** in decoder layers that are most responsible for certain safety behaviors under the original model parameters. In contrast, JDJN identifies **MLP neurons** that support the counterfactual refusal behavior under jailbreak prompts. These two perspectives (attention heads vs. MLP; original vs. counterfactual behavior) are complementary. Extending our framework to “jailbreak heads” and connecting them with the “JailNeurons” we identify is an interesting direction for future work.
> >
> > -  **[6] HiddenDetect.**
> >
> > HiddenDetect uses logit‑lens decoding to interpret intermediate representations, which tends to be less reliable for shallow layers. As a result, it under‑uses early‑layer information. JDJN, guided by the explicit “sure → sorry” objective, selects JailNeurons across **all depths**, including shallow layers, and then trains a simple classifier on their activations. This allows JDJN to leverage both low‑level and high‑level signals for more comprehensive detection.
> >
> > To make these differences more concrete, we add the following comparison (all use TPR@FPR ≤ 0.05, detector trained on JailBreakV + MM‑Vet with FPR ≤ 0.05 on MM‑Vet):
> >
> > | Method| JailBreakV (LLaVA) | FigStep (LLaVA) | JAMLLM (LLaVA) | JailBreakV (Janus‑Pro) | FigStep (Janus‑Pro) | JAMLLM (Janus‑Pro)|
> > |:------:|:------:|:------:|:------:|:--------:|:------:|:------:|
> > | **JDJN** | 0.997 | 1.000 | 0.942 | 0.996 | 1.000 | 0.853 |
> > | First‑to‑Know | 1.000 | 0.952 | 0.433 | 1.000 | 0.976 | 0.323 |
> > | SNIP | 0.993 | 0.995 | 0.896 | 0.996 | 0.932 | 0.623 |
> > | HiddenDetect | 0.335 | 0.552 | 0.340 | 0.415 | 0.624 | 0.611 |
> >
> > **Table 1.** Comparison of TPR@FPR ≤ 0.05 for different neuron‑digging–based methods.
> >
> > We observe that JDJN achieves the best overall generalization, particularly on out‑of‑distribution datasets such as FigStep and JAMLLM.

---

> > > ### Author Response · Authors · 2025-11-21
> > > **Author Response (part 3 of 3)**
> > >
> > > ### 3. Additional experiments on OR‑Bench and XSTest (false positives and over‑sensitivity)
> > >
> > >
> > >
> > > **Our response.**
> > >
> > > We thank the reviewer for emphasizing the importance of false positives and generalization. Following this suggestion, we additionally evaluate JDJN on two benign MLLM datasets and two textual stress‑test benchmarks:
> > >
> > >
> > >
> > > -  **ScreenSpots:** grounding task benchmark for LVLMs. [7]
> > >
> > > -  **AndroidControl:** GUI agent benchmark for LVLMs. [8]
> > >
> > > -  **OR‑Bench** [9] and **XSTest:** [10] stress‑test benchmarks designed to measure **over‑refusal** in LLMs.
> > >
> > > We focus on JDJN$_1$​, which is trained on JailBreakV and MM‑Vet with FPR ≤ 0.05 on MM‑Vet. We then test its FPR on multiple benign datasets and stress tests:
> > >
> > >
> > > |Model|MM‑Vet|MM‑Bench|Normal|ScreenSpots|AndroidControl|OR‑Bench|XSTest|
> > > |:------:|:------:|:------:|:------:|:--------:|:------:|:------:|:------:|
> > > |LLaVA|0.000|0.000|0.019|0.022|0.012|0.390|0.170|
> > > |Janus‑Pro|0.000|0.000|0.044|0.002|0.033|0.280|0.090
> > >
> > > **Table 2.** FPR of JDJN on benign and stress‑test datasets.
> > >
> > > We make two observations:
> > >
> > > 1. On **benign distributions** that more closely resemble normal user scenarios (e.g., ScreenSpots, AndroidControl, and our Normal / MM‑Bench subsets), JDJN maintains a **low FPR (≤ 0.05)**, indicating that it is not overly sensitive in typical use cases.
> > >
> > > 2. JDJN exhibits relatively higher FPR on **OR‑Bench** and **XSTest**.
> > >
> > > However, we argue that these datasets are explicitly constructed as **stress tests for over‑refusal** and are known to be challenging even for commercial models (e.g., GPT‑4, Gemini), which show over‑refusal rates exceeding 90% on OR‑Bench [9]. Our FPR on these benchmarks is therefore high in absolute terms but still moderate compared with the over‑refusal rates observed in aligned LLMs.
> > >
> > > Moreover, our design goal in constraining FPR ≤ 0.05 during training is to avoid harming normal user experience on typical benign inputs. Since OR‑Bench and XSTest intentionally target extreme corner cases of refusal behavior, they are less representative of the majority of benign user traffic. Thus, while JDJN is stricter on these stress‑test prompts, its behavior on ordinary benign inputs remains conservative and practical.
> > >
> > >
> > > ----------
> > >
> > >
> > > ### 4. Clarity of figures and tables
> > >
> > > **Our response.**
> > >
> > > We appreciate this comment and agree that clearer figures and captions are important for readability. In the revised version, we:
> > >
> > >
> > > - Explicitly state in the main text and captions that the default metric is **TPR@FPR ≤ 0.05**, unless otherwise specified.
> > >
> > > - Clarify that **detection rate** is defined as
> > >
> > > $TPR= \frac{\text{ ||successfully detected jailbreak samples||}}{||\text{ all (successful) jailbreak samples}||}$
> > >
> > >
> > >
> > > ----------
> > >
> > > **Once again, we thank the reviewer for the insightful comments and for highlighting both the strengths (clear writing, interesting optimization‑based neuron selection) and areas for improvement (clearer comparison to prior neuron‑digging works, detailed baseline setup, and additional evaluations). We believe the suggested clarifications and new experiments substantially strengthen the manuscript.**
> > >
> > > -----
> > >
> > > [1] Two Souls in an Adversarial Image: Towards Universal Adversarial Example Detection using Multi-view Inconsistency
> > >
> > > [2] Jailbreaking Attack against Multimodal Large Language Model
> > >
> > > [3] The First to Know: How Token Distributions Reveal Hidden Knowledge in Large Vision-Language Models?
> > >
> > > [4] Assessing the Brittleness of Safety Alignment via Pruning and Low-Rank Modifications.
> > >
> > > [5] On the Role of Attention Heads in Large Language Model Safety
> > >
> > > [6] HiddenDetect: Detecting Jailbreak Attacks against Large Vision-Language Models via Monitoring Hidden States
> > >
> > > [7] SeeClick: Harnessing GUI Grounding for Advanced Visual GUI Agents
> > >
> > > [8]On the Effects of Data Scale on UI Control Agents
> > >
> > > [9] OR-Bench: An Over-Refusal Benchmark for Large Language Models
> > >
> > > [10] XSTest: A Test Suite for Identifying Exaggerated Safety Behaviours in Large Language Models

---

> ### Author Response · Authors · 2025-11-26
>
> Dear Reviewer pMai,
>
> As the discussion phase enters its final week, we kindly ask if you could review our response to ensure it addresses your concerns. If you have any further questions, please feel free to let us know. Your feedback is greatly appreciated.
>
> Thank you for your time!
>
> Best,
>
> Authors

---

> > ### Comment · Reviewer_pMai · 2025-11-26
> > **Thanks for the rebuttal.**
> >
> > Thanks for the detailed rebuttal. Most of my concerns are settled. It is encouraged to add these comparisons, as well as the detailed experimental setups in the script. Again, I appreciate the time authors devote to the discussion, and I will increase my score to 6.

---

> > > ### Author Response · Authors · 2025-11-27
> > >
> > > Thank you very much for your considerate response and for increasing your score. We sincerely appreciate the time and effort you invested in reviewing our work and engaging in the discussion.
> > >
> > > Following your suggestions, we have updated the manuscript. In particular, we included the additional comparisons and provided more detailed descriptions of the experimental setups, mainly in Section 5.1, Section 7.1 and Section 7.3 with blue-highlighted text. If you have any further questions or comments, we would be more than happy to address them.

---

### Official Review · Reviewer_GpcJ · 2025-10-29

**Soundness:** 2
**Presentation:** 3
**Contribution:** 2
**Rating:** 4
**Confidence:** 4

**Summary:**

This paper introduces JDJN (Jailbreak Detection via JailNeurons), a novel method for detecting jailbreak attacks in Large Vision-Language Models (LVLMs). JDJN introduces the concept of "JailNeurons" - specific neurons that are activated during jailbreak attempts
These neurons are distinct from previously studied "SafeNeurons" which explain standard safety mechanisms.

**Strengths:**

This paper introduces JDJN, a novel method for detecting jailbreak attacks in Large Vision-Language Models by identifying and leveraging "JailNeurons". The approach demonstrates strong originality in its conceptualization of JailNeurons and its creative "sure-to-sorry" localization procedure. The quality of the work is evidenced by comprehensive empirical validation across multiple models and attack types, achieving impressive detection rates while maintaining computational efficiency. The significance of this research is substantial, addressing a critical security challenge in LVLMs with a practical, generalizable solution that could have immediate real-world impact on improving AI system safety.

**Weaknesses:**

1. Section 4.2.1 introduces a mask which is the key to this work. I am wondering whether this mask is neccessary. If the jailbreak information is in the neuron, why we cannot learn a classifier directly? If there is need of filtering out unrelated neurons, you have different options like regularizations and etc.

2. You are missing some baselines in Table1. For example, AdaShield and JailDAM and Gradsafe and etc.

3. For different dataset, how will the mask changing? It will be interesting to know how this changes. If the mask is different for different dataset, how do you explain the neuron you find?

**Questions:**

See weaknesses.

---

> ### Author Response · Authors · 2025-11-21
> **Author Response (part 1 of 3)**
>
> ### 1. On the necessity of the mask in Section 4.2.1
>
> **Our response**
>
> We thank the reviewer for this insightful question about the role and necessity of the mask.
>
> We design the JailNeurons mask as a neuron‑level filtering step before training the detector, and we find it both conceptually meaningful and empirically important. We provide justification from both **intuition** and **experimental** perspectives.
>
> **Intuitive justification.**
>
> Without any filtering, the neuron activations are extremely high‑dimensional. For example, Qwen‑VL‑8B has 4,096 neurons per layer, and even 10 layers already yield more than 40,000 activation dimensions per input. With only a few hundred training samples, directly learning a robust classifier on such a space easily suffers from overfitting and the curse of dimensionality.
>
> If we want to reduce dimensionality or filter irrelevant neurons, several alternative strategies are possible:
>
> 1. **JailNeurons‑based filtering (ours):** selects neurons that respond specifically to jailbreak transitions (the “sure‑to‑sorry” behavior), directly highlighting internal representations most related to jailbreak behavior.
> 2. **SNIP‑based filtering:** identifies safety‑related neurons instead of neurons tied to jailbreak behavior, making it less targeted for the jailbreak detection task.
> 3. **Generic dimensionality reduction (e.g., PCA):** compresses features according to variance, without alignment to safety or jailbreak semantics.
> 4. **Generic regularization (e.g., $L_1$, $L_2$​):** training the SVM with regularization helps to some extent, but still operates on a very large, mostly unstructured feature space.
>
> Our design explicitly targets neurons correlated with jailbreak semantics, rather than generic harmfulness or arbitrary high‑variance directions. This yields a more meaningful and compact feature set for the downstream classifier.
>
> **Empirical justification.**
>
> To verify the necessity of the mask, we conduct a controlled comparison of six variants on LLaVA, all trained on JailBreakV and MM‑Vet:
>
> 1. **JDJN (ours):** JailNeurons filtering + top‑down layer sampling + SVM.
> 2. **No filter, no regularization (NFNR):** all neuron activations + top‑down sampling + SVM.
> 3. **No filter, $L_1$​ regularization:** all neuron activations + top‑down sampling + SVM with $L_1$​.
> 4. **No filter, $L_2$​ regularization:** all neuron activations + top‑down sampling + SVM with $L_2$.
> 5. **PCA‑based filtering:** PCA in place of JailNeurons + top‑down sampling + SVM.
> 6. **SNIP‑based filtering:** SNIP‑selected neurons in place of JailNeurons + top‑down sampling + SVM.
>
> |Method|JailBreakV (TPR)|FigStep (TPR)|JAMLLM (TPR)|MM‑Bench (FPR)|MM‑Vet (FPR)|Normal (FPR)|
> | :----: |:----: |:----: |:----: |:----: |:----: |:----: |
> |**JDJN**|**0.997**|**1.000**|**0.942**|**0.000**|**0.000**|**0.019**|
> |NFNR|0.993|1.000|0.874|0.000|0.075|0.472|
> |L1​ regu|1.000|1.000|0.734|0.000|0.052|0.246|
> |L2​ regu|1.000|0.996|0.812|0.000|0.000|0.218|
> |PCA|0.981|1.000|0.775|0.005|0.010|0.626|
> |SNIP|0.993|0.998|0.896|0.000|0.086|0.577|
>
> We observe that JDJN consistently achieves the best trade‑off: high TPR on jailbreak datasets and low FPR on benign datasets. In particular, on out‑of‑distribution benign data (Normal), JDJN reduces FPR from 0.442 (no filter) to 0.019. Methods without JailNeurons‑based masking, even with regularization or PCA, either overfit (high FPR) or underperform on TPR.
>
> These results support that the JailNeurons mask is not only a design choice but also a practically important component that isolates jailbreak‑specific features more effectively than naïve dimensionality reduction or generic regularization. We will clarify this motivation and add the above comparison in the revised version.

---

> > ### Author Response · Authors · 2025-11-21
> > **Author Response (part 2 of 3)**
> >
> > ### 2. On mask variability across datasets
> >
> > **Our response**
> >
> > We appreciate this question on the generalizability and interpretability of the learned masks, and we agree that understanding how JailNeurons vary across datasets is important.
> >
> > Our working hypothesis is that jailbreak samples bypass the model’s defenses by activating specific neurons, which we call JailNeurons. Different jailbreak distributions (e.g., different attack methods or intents) may activate partially different sets of neurons, but we expect substantial overlap in neurons that encode shared jailbreak patterns.
> >
> > **Overlap of JailNeurons across jailbreak datasets**
> >
> > To analyze this, we compute the overlap between JailNeurons identified from three different jailbreak datasets: JailBreakV, FigStep, and JAMLLM. For each jailbreak method $i \in$ &lcub; JailBreakV, FigStep, JAMLLM &rcub;, we denote its neuron set as $J_i$. For any pair $(i,j)$, we measure:
> >
> > $$
> > p_{ij} = \frac{||  J_i \bigcap J_j  ||  }{ ||J_i|| }​
> > $$
> >
> > i.e., the proportion of JailNeurons from method i that also appear in method j.
> >
> > The overlaps are:
> >
> > ||JailBreakV (LLaVA)|FigStep (LLaVA)|JAMLLM (LLaVA)|JailBreakV (Janus‑Pro)|FigStep (Janus‑Pro)|JAMLLM (Janus‑Pro)|
> > | :----: |:----: |:----: |:----: |:----: |:----: |:----: |
> > |**JailBreakV**|1.00|0.52|0.31|1.00|0.68|0.28|
> > |**FigStep**|0.96|1.00|0.26|0.98|1.00|0.18|
> > |**JAMLLM**|0.48|0.12|1.00|0.43|0.16|1.00|
> >
> > We observe that:
> >
> > - JailBreakV and FigStep share a large portion of JailNeurons (e.g., FigStep neurons are almost a subset of JailBreakV neurons). This is consistent with the fact that some JailBreakV prompts come from FigStep.
> > - There is also a non‑trivial overlap between JAMLLM and JailBreakV (around 30–40%), suggesting that different jailbreak benchmarks still share common internal patterns.
> >
> > This supports the view that JailNeurons consist of:
> >
> > (i) **shared components** capturing generic jailbreak mechanisms, and
> >
> > (ii) **dataset‑specific components** capturing characteristics unique to each attack distribution.
> >
> > ** Effect of different parts of JailNeurons**
> >
> > To further interpret these neurons, we focus on JAMLLM and JailBreakV as a case study and partition the JailNeurons into three disjoint subsets:
> >
> > - $J_{\text{JAMLLM}}​$: neurons unique to JAMLLM,
> > - $J_{\text{JailBreakV}}​$: neurons unique to JailBreakV,
> > - $J_{\text{overlap}}$​: neurons shared by both.
> >
> > We also use a random baseline, $J_{\text{random}}$​, where we randomly deactivate the same number of neurons as in $J_{\text{JailBreakV}}​$.
> >
> > We then deactivate each subset and measure the confidence that LLaVA outputs “Sorry” on each dataset. Let $P_{\text{LLaVA}}(\text{Sorry} \mid X)$ denote the “Sorry” confidence on dataset $X \in \{\text{JAMLLM}, \text{JailBreakV}\}$.
> >
> > **The values of $P_{\text{LLaVA}}(\text{Sorry} \mid JAMLLM)$:**
> >
> > |Deactivated neurons|Layer 1|Layer 5|Layer 9|Layer 13|Layer 17|Layer 21|Layer 25|
> > | :----: |:----: |:----: |:----: |:----: |:----: |:----: |:----: |
> > |$J_ {\text{JAMLLM}}$​|0.377|0.443|0.412|0.396|0.376|0.357|0.354|
> > |$J_{\text{JailBreakV}}$​|0.194|0.303|0.276|0.202|0.185|0.183|0.186|
> > |$J_{\text{overlap}}$​|0.325|0.415|0.417|0.402|0.354|0.320|0.312|
> > |$J_{\text{random}}$​|≤ 0.001|≤ 0.001|≤ 0.001|≤ 0.001|≤ 0.001|≤ 0.001|≤ 0.001|
> >
> > **The values of $P_{\text{LLaVA}}(\text{Sorry} \mid JailBreakV)$:**
> >
> > |Deactivated neurons|Layer 1|Layer 5|Layer 9|Layer 13|Layer 17|Layer 21|Layer 25|
> > | :----: |:----: |:----: |:----: |:----: |:----: |:----: |:----: |
> > |$J_{\text{JAMLLM}}$​|0.092|0.113|0.073|0.057|0.059|0.055|0.042|
> > |$J_{\text{JailBreakV}}$​|0.284|0.302|0.176|0.196|0.182|0.183|0.177|
> > |$J_{\text{overlap}}$​|0.243|0.276|0.114|0.098|0.103|0.109|0.087|
> > |$J_{\text{random}}$​|≤ 0.001|≤ 0.001|≤ 0.001|≤ 0.001|≤ 0.001|≤ 0.001|≤ 0.001|
> >
> > We find that:
> >
> > - Deactivating **JAMLLM‑specific** neurons or **overlap** neurons significantly increases the “Sorry” confidence on JAMLLM.
> > - Deactivating **JailBreakV‑specific** neurons or **overlap** neurons significantly increases the “Sorry” confidence on JailBreakV.
> > - All three learned subsets are far more influential than deactivating **random neurons**.
> >
> > These results suggest that:
> >
> > - The **overlap part** $J_{\text{overlap}}$​ captures core jailbreak behavior shared across datasets.
> > - The **dataset‑specific parts** $J_{\text{JAMLLM}}​$ and $J_{\text{JailBreakV}}$​ still capture signals that can generalize partially to other jailbreak distributions.
> >
> > Even though we impose an $L_1$​ constraint to limit mask size and filter out neurons less relevant to the training distribution, the resulting masks still retain rich jailbreak‑related information that **transfers across attack types**. We incorporate this analysis and a clearer discussion about shared vs. dataset‑specific JailNeurons in the revised paper.

---

> ### Author Response · Authors · 2025-11-21
> **Author Response (part 3 of 3)**
>
> ### 3. On missing baselines
>
> **Our response**
>
> We thank the reviewer for highlighting the importance of including more recent baselines. We add AdaShield, JailDAM, and GradSafe to our main comparison and report TPR@FPR=0.05 for both LLaVA and Janus‑Pro. (Since AdaShield is not a detection algorithm, we consider that the output of jailbreak samples after AdaShield defense, which includes a refusal token, indicates a successful detection.)
>
> Method|JailBreakV (LLaVA)|FigStep (LLaVA)|JAMLLM (LLaVA)|JailBreakV (Janus‑Pro)|FigStep (Janus‑Pro)|JAMLLM (Janus‑Pro)|
> | :----: |:----: |:----: |:----: |:----: |:----: |:----: |
> |**JDJN**|**0.997**|**1.000**|**0.942**|**0.996**|**1.000**|**0.853**|
> |GradSafe|0.862|0.742|0.534|0.844|0.728|0.454|
> |JailDAM|0.913|0.926|0.342|0.917|0.932|0.433|
> |AdaShield|0.675|0.786|0.213|0.774|0.812|0.353|
>
> Across both models and all three attack benchmarks, JDJN achieves the highest TPR at the same FPR level, which indicates that our neuron‑based approach is competitive with, and often substantially stronger than, existing defense mechanisms.
>
> ----
>
> **Once again, we thank the reviewer for the insightful comments and for highlighting both the strengths (strong originality in conceptualization of JailNeurons and impressive detection rates) and areas for improvement (necessity of the mask, more baselines and mask variability across datasets). We believe the suggested clarifications and new experiments substantially strengthen the manuscript.**

---

> ### Author Response · Authors · 2025-11-26
>
> Dear Reviewer GpcJ,
>
> As the discussion phase enters its final week, we kindly ask if you could review our response to ensure it addresses your concerns. If you have any further questions, please feel free to let us know. Your feedback is greatly appreciated.
>
> Thank you for your time!
>
> Best,
>
> Authors

---

### Official Review · Reviewer_dJcj · 2025-10-30

**Soundness:** 2
**Presentation:** 3
**Contribution:** 3
**Rating:** 6
**Confidence:** 3

**Summary:**

This paper introduces JDJN (Jailbreak Detection via JailNeurons), a framework for detecting jailbreak attacks in LVLMs by identifying and aggregating neuron activations that are responsible for jailbreak behavior. The method 1. localizes a sparse set of “JailNeurons” through a learned masking optimization 2. trains a lightweight classifier (linear SVM) over these neurons across selected layers

**Strengths:**

1. Experiments on four LVLMs and three attack types show high true-positive rates.
2. The proposed method seems to have strong generalization ability.

**Weaknesses:**

1. The “causal” interpretation of Eq. (2–3) is asserted but not formally proven. Only intervention -> change is shown but no contrastive group of neurons is shown to not affect the output.
2. Performance drops on certain benign datasets (e.g., FPR 0.768 on Janus-pro/Normal in Table 3). The parameter settings (JDJN3 in this case) seem to impact the generalization of the proposed approach.
3 . Efficiency analysis is not very comprehensive. The authors claim that JDJN requires only a single forward pass but the details on the mask localization and training of the classifier is missing.
4. Potential overfitting to limited benchmarks.

**Questions:**

See weakness.
1. It would help if the authors can provide more rigorous experiment/ analysis on the chosen jailneurons.
2. Can the authors provide more discussion on the failure cases.
3. Can the authors provide a more comprehensive analysis on the proposed method? For example, what would be the mask localization cost and classifier training cost?

---

> ### Author Response · Authors · 2025-11-21
> **Author Response (part 1 of 3)**
>
> ### 1. On the causal interpretation of JailNeurons
>
> **Our response**
>
> We thank the reviewer for raising this important point about causal interpretation. To directly probe the causal role of JailNeurons, we conduct a causal‑contrastive analysis by deactivating different neuron sets and measuring the change in refusal probability.
>
> Concretely, for each layer, we:
>
> - deactivate the JailNeurons identified by JDJN, and
>
> - compare this to randomly deactivating neurons.
>
> We denote:
>
>
> -  **RandNeurons1**: randomly removing the same number of neurons as JailNeurons;
>
>
> -  **RandNeurons5**: randomly removing five times the number of JailNeurons.
>
>
> We randomly sample 500 jailbreak examples from JailBreak‑V that successfully attack the original model and report the average confidence that the model outputs _“Sorry”_ under three conditions: deactivating JailNeurons, deactivating random neurons, and no modification.
>
>
> **Janus‑Pro:**
>
> |masked_layer_id |1 |5|9|13|17|21|25|
> | :----: |:----: |:----: |:----: |:----: |:----: |:----: |:----: |
> |JailNeurons|0.208|0.205|0.204|0.206|0.202|0.202|0.202|
> |RandNeurons1|<0.001|<0.001|<0.001|<0.001|<0.001|<0.001|<0.001|
> |RandNeurons5|0.001|0.001|0.001|0.001|0.001|<0.001|0.001|
> |No mask|<0.001| <0.001|<0.001|<0.001|<0.001|<0.001|<0.001|
>
>
> **LLaVA:**
>
> |masked_layer_id |1 |5|9|13|17|21|25|
> | :----: |:----: |:----: |:----: |:----: |:----: |:----: |:----: |
> |JailNeurons|0.393|0.457|0.260|0.278|0.260|0.249|0.279|
> |RandNeurons1|≤0.001|≤0.001|≤0.001|≤0.001|≤0.001|≤0.001|≤0.001|
> |RandNeurons5|0.001|0.002|0.005|0.002|0.002|0.001|0.001|
> |No mask|≤0.001|≤0.001|≤0.001|≤0.001|≤0.001|≤0.001|≤0.001|
>
> We observe that:
>
> - Without any intervention, the original models output _“Sorry”_ with extremely low probability on successful jailbreak samples.
>
> - When we deactivate JailNeurons, the probability of _“Sorry”_ sharply increases.
>
> - When we randomly deactivate neurons (even at 5× the count of JailNeurons), the _“Sorry”_ probability remains near zero.
>
> This causal‑contrastive gap indicates that JailNeurons are not just arbitrary high‑activation units, but are specifically and causally linked to the model’s safety mechanisms that are being bypassed during jailbreak. This provides stronger support for a causal interpretation than purely correlational evidence.
>
> ----------
>
> ### 2. On the high FPR of JDJN$_3$ on the Normal dataset
>
>
> **Our response**
>
> We appreciate the reviewer’s careful inspection of the false positive rates.
>
> JDJN$_3$ exhibits a higher FPR on the Normal dataset compared to JDJN$_1$. However, this behavior mainly stems from the characteristics of the benign training data, rather than from different parameter settings, since JDJN$_3$ and JDJN$_1$ share identical hyperparameters.
>
> JDJN performs detection via the trained SVM. Like most machine learning models, the quality and diversity of the training data are critical. MM‑Bench, which is used as benign data for JDJN$_3$, is relatively constrained: it restricts the model outputs to multiple‑choice options [A,B,C,D.] This introduces a strong shortcut pattern. As a result, even though JailNeurons filter some neurons, the SVM trained under the JDJN$_3$ setting cannot completely remove the spurious correlations induced by this specific output format.
>
> By contrast, when we train JDJN$_1$ using MM‑Vet and JailBreakV, the resulting detector generalizes well to MM‑Bench and Normal. This indicates that JDJN as a framework has strong potential as a general‑purpose jailbreak detector, and that its robustness is sensitive to the diversity of benign training data. We will highlight this data‑dependence more explicitly and clarify in the paper that the high FPR of JDJN$_3$ is largely due to MM‑Bench’s particular format.

---

> > ### Author Response · Authors · 2025-11-21
> > **Author Response (part 2 of 3)**
> >
> > ### 3. On potential overfitting to current benchmarks
> >
> > In response, we extend our evaluation to include two additional benign datasets:
> >
> >
> > -  **ScreenSpots**: assesses LVLM performance on grounding tasks [1];
> >
> > -  **AndroidControl**: assesses LVLM performance on GUI agent control tasks [2].
> >
> > These datasets cover different benign usage patterns and interaction styles, thereby offering a more diverse set of non‑jailbroken inputs.
> >
> >
> > We evaluate JDJN$_1$​ (trained on JailBreak‑V and MM‑Vet) on these new benign datasets. The false positive rates (FPR) are as follows:
> >
> > |Model|MM‑Vet|MM‑Bench|Normal|ScreenSpots|AndroidControl|
> > | :----: |:----: |:----: |:----: |:----: |:----: |
> > | LLaVA |0.000|0.000|0.019|0.022|0.012|
> > |Janus‑Pro|0.000|0.000|0.044|0.002|0.033|
> >
> > Across all five benign datasets, JDJN maintains low FPR (≤ 0.05). This expanded evaluation supports the stability of JDJN1_11​ under broader benign distributions and suggests good potential for realistic deployment scenarios where benign inputs are diverse and task‑specific.
> >
> > ----------
> >
> > ### 4. On efficiency and computational cost
> >
> > **Our response**
> >
> > We appreciate the reviewer’s request for a more complete efficiency analysis and provide further details here.
> >
> > Training JDJN consists of two main stages:
> >
> >
> > 1.  **JailNeurons localization**, which learns sparse masks over neurons.
> >
> > 2.  **SVM training**, which learns a lightweight classifier over the selected neuron activations.
> >
> > Let the number of jailbreak training samples be nnn, the number of optimization steps per sample be sss, the number of targeted key layers be kkk, and the batch size be bbb. The overall training cost is approximately $\frac{n \times s \times k}{b}$ forward–backward passes.
> >
> > In our experiments (for LLaVA), we set: n=400, s=100,  k=6 and b=1.
> >
> > Under this configuration, the JailNeurons localization stage takes about **30 hours** on a single A800 GPU. After localization, we extract neuron activations and train a **linear SVM**, which completes within **tens of seconds**, due to the low dimensionality and modest data size.
> >
> > Thus, the main cost lies in the **one‑time** JailNeurons localization. Once JailNeurons and the SVM are trained, **inference** for a new input requires only:
> >
> >
> > - a **single forward pass** through the LVLM to obtain neuron activations, and
> >
> > - a negligible SVM evaluation.
> >
> >
> > We will add these quantitative details and explicitly distinguish between one‑time training cost and per‑sample inference cost in the revised paper to avoid any ambiguity about the efficiency claim.

---

> > > ### Author Response · Authors · 2025-11-21
> > > **Author Response (part 3 of 3)**
> > >
> > > ### 5. On the analysis of failure cases
> > >
> > > **Our response**
> > >
> > > We agree that analyzing failure modes is essential for understanding the limitations of JDJN, and we thank the reviewer for encouraging a deeper discussion.
> > >
> > > By examining misclassified examples, we observe that most errors arise for prompts that are **semantically borderline** between benign and malicious:
> > >
> > > -  **False positives.**
> > >
> > > On the Normal dataset, an example such as _“Please list key events from World War II”_ is misclassified as a jailbreak sample. While this request is benign in intent, it contains war‑related concepts that may resemble patterns present in some malicious jailbreak prompts, leading JDJN to over‑flag it.
> > >
> > > -  **False negatives.**
> > >
> > > On JAMLLM, we observe cases where a prompt is labeled as malicious in the benchmark, but the model’s actual response is neutral and contains minor malicious information. Because the model does not explicitly reject the prompt with a safety message, the we label the interaction as a jailbreak, whereas JDJN classifies it as benign.
> > >
> > > These observations suggest that:
> > >
> > > 1. JDJN tends to be sensitive to the presence of certain **sensitive topics or concepts**, even when the intent is benign, which can cause false positives; and
> > >
> > > 2. A mismatch between **dataset labeling criteria** and **model‑behavioral safety** can create apparent false negatives, especially for prompts that are not explicitly rejected but also not answered harmfully.
> > >
> > > We will add these representative examples and a more systematic discussion of failure patterns in the revision, and we see this as a promising direction for future work on refining both the detector and evaluation protocols.
> > >
> > > ----
> > >
> > > **Once again, we thank the reviewer for the insightful comments and for highlighting both the strengths (strong generalization ability and high true-positive rates) and areas for improvement (causal  interpretation, efficiency analysis and limited benchmarks). We believe the suggested clarifications and new experiments substantially strengthen the manuscript.**
> > >
> > > ----
> > >
> > > [1] SeeClick: Harnessing GUI Grounding for Advanced Visual GUI Agents
> > >
> > > [2] On the Effects of Data Scale on UI Control Agents

---

> > > > ### Author Response · Authors · 2025-11-26
> > > >
> > > > Dear Reviewer dJcj,
> > > >
> > > > As the discussion phase enters its final week, we kindly ask if you could review our response to ensure it addresses your concerns. If you have any further questions, please feel free to let us know. Your feedback is greatly appreciated.
> > > >
> > > > Thank you for your time!
> > > >
> > > > Best,
> > > >
> > > > Authors

---

### Official Review · Reviewer_5qV5 · 2025-10-31

**Soundness:** 3
**Presentation:** 3
**Contribution:** 2
**Rating:** 6
**Confidence:** 4

**Summary:**

This paper studies jailbreak detection for Large Vision-Language Models (LVLMs) by identifying a small set of neurons that strongly relate to jailbreak behavior. The authors first show that jailbreak and benign samples trigger different internal activations but that naive linear detection does not generalize well across attack types and benign sources. They then introduce JDJN, which locates “JailNeurons” through a mask-based optimization that forces harmful outputs to switch to refusal responses, and aggregates activations from selected layers to train a lightweight detector. Experiments across several LVLMs, three attack types, and multiple benign datasets indicate that JDJN yields high true-positive rates with very low false-positive rates, and generalizes to out-of-distribution attacks and unseen benign distributions. The method is efficient and works without modifying the base LVLM, making deployment friendly.

**Strengths:**

The paper frames a clear and specific safety question: whether jailbreak activity concentrates in a sparse set of neurons and whether such neurons can support consistent, low-cost detection. This goes beyond generic claims like “performance limits” and targets internal mechanisms of LVLM jailbreak behavior. The method is well-aligned with that motivation, since the mask-training step is crafted to pinpoint neurons whose ablation flips harmful responses to safe refusals. Both single-layer and multi-layer analyses are thorough, and the adversarial evaluation with adaptive attacks adds credibility. The empirical results demonstrate strong evidence rather than speculation, including cross-model tests and ablation studies that examine critical components such as layer selection, mask threshold, regularization, and detector choice. Overall, the study links motivation, method, and experiments coherently, and offers a useful tool for practical safety settings.

**Weaknesses:**

The causal interpretation of “JailNeurons” could be more rigorous; while the mask-based optimization offers a constructive handle, the paper does not fully rule out the possibility that the selected neurons encode surface-level shortcuts tied to particular phrasing or datasets.

Although the authors run OOD tests, the scope of benign distributions is still somewhat narrow, and the stability of neuron sets across architectures and scaling regimes could benefit from deeper analysis.

There is also limited exploration of joint vision-language pathways; the focus is on text-side activations, so multimodal interplay is not fully dissected.

The white-box assumption may restrict real-world deployment scenarios, and a discussion on extending the method toward limited-access or proxy-signal settings would strengthen the broader impact.

**Questions:**

You report that JailNeurons are sparse and effective. Do these neurons remain consistent under model finetuning, safety tuning, or instruction-following updates? How stable are they across different checkpoints of the same LVLM architecture?

The paper mainly analyzes decoder-side neurons. Do convolutional / vision transformer layers contain JailNeurons as well? If so, is the jailbreak signal similarly sparse? If not, what does that imply about multimodal interaction during jailbreak?

---

> ### Author Response · Authors · 2025-11-21
> **Author Response (part 1 of 3)**
>
> ## 1. On the causal validity of JailNeurons and potential surface‑level shortcuts
> **Our response.**
> We thank the reviewer for raising this important point about causal interpretation. We clarify and strengthen our position with two complementary pieces of evidence that support a causal link between JailNeurons and jailbreak behavior, beyond surface‑level correlations.
> ### (i) Evidence from cross‑dataset generalization
> Our six datasets cover diverse image types and text descriptions, with substantial distribution shifts across datasets. If JDJN only captures surface‑level shortcuts tied to specific phrasing patterns or dataset artifacts, it is unlikely to generalize well across such heterogeneous sources.
> However, JDJN consistently maintains strong out‑of‑distribution (OOD) performance across different attack types and benign datasets. This stable generalization behavior supports the view that JDJN captures deeper causal semantics related to jailbreak behavior, rather than simply memorizing surface regularities.
>
> ### (ii) Causal‑contrastive validation via neuron deactivation
>
> To more directly probe the causal role of JailNeurons, we conduct a causal‑contrastive analysis by deactivating different neuron sets and measuring the change in refusal probability.
>
> Concretely, for each layer, we:
>
> - deactivate the JailNeurons identified by JDJN, and
>
> - compare this to randomly deactivating neurons.
>
> We denote:
>
> -  **RandNeurons1**: randomly removing the same number of neurons as JailNeurons;
>
> -  **RandNeurons5**: randomly removing five times the number of JailNeurons.
>
> We randomly sample 500 jailbreak examples from JailBreak‑V that successfully attack the original model and report the **average confidence** that the model outputs _“Sorry”_ under three conditions: deactivating JailNeurons, deactivating random neurons, and no modification.
>
> **Janus‑Pro:**
>
> |masked_layer_id |1 |5|9|13|17|21|25|
> | :----: |:----: |:----: |:----: |:----: |:----: |:----: |:----: |
> |JailNeurons|0.208|0.205|0.204|0.206|0.202|0.202|0.202|
> |RandNeurons1|<0.001|<0.001|<0.001|<0.001|<0.001|<0.001|<0.001|
> |RandNeurons5|0.001|0.001|0.001|0.001|0.001|<0.001|0.001|
> |No mask|<0.001| <0.001|<0.001|<0.001|<0.001|<0.001|<0.001|
>
> **LLaVA:**
>
> |masked_layer_id |1 |5|9|13|17|21|25|
> | :----: |:----: |:----: |:----: |:----: |:----: |:----: |:----: |
> |JailNeurons|0.393|0.457|0.260|0.278|0.260|0.249|0.279|
> |RandNeurons1|<0.001|<0.001|<0.001|<0.001|<0.001|<0.001|<0.001|
> |RandNeurons5|0.001|0.002|0.005|0.002|0.002|0.001|0.001|
> |No mask|<0.001|<0.001|<0.001|<0.001|<0.001|<0.001|<0.001|
>
> We observe that:
> - Without any intervention, the original models output _“Sorry”_ with extremely low confidence on successful jailbreak samples.
> - When we deactivate JailNeurons, the confidence of _“Sorry”_ sharply increases.
> - When we randomly deactivate neurons (even at 5× the count of JailNeurons), the _“Sorry”_ probability remains near zero.
>
> This causal‑contrastive gap indicates that JailNeurons are not just arbitrary high‑activation units, but are specifically and causally linked to the model’s safety mechanisms that are being bypassed during jailbreak. This provides stronger support for a causal interpretation than purely correlational evidence.
>
> ## 2. On JailNeurons in the vision module
>
>
>
>
> **Our response.**
>
> We thank the reviewer for pointing out this important multimodal aspect. In response, we extend our analysis to the vision backbone (ViT) and apply our neuron importance estimation there.
>
>
>
> ### Sparsity of visual JailNeurons
>
>
>
> We compute the proportion of JailNeurons in several layers of the LLaVA vision module:
>
>
>
> |Layer id|2|5|8|11|14|17|20|
> | :----: |:----: |:----: |:----: |:----: |:----: |:----: |:----: |
> |Proportion (%)|0.97|1.26|0.97|0.39|0.39|0.39|0.68|
>
>
>
> The proportion of JailNeurons in the vision module remains below 1% across these layers, indicating that jailbreak‑relevant signals in the visual pathway are also sparse.
>
>
>
> ### Functional impact of deactivating visual JailNeurons
>
>
>
> Analogous to the text‑side analysis, we measure the model’s confidence of outputting _“Sorry”_ after deactivating the identified visual JailNeurons:
>
>
>
> |Layer id|2|5|8|11|14|17|20|
> | :----: |:----: |:----: |:----: |:----: |:----: |:----: |:----: |
> |Confidence|0.012|0.017|0.016|0.005|0.005|0.003|0.011|
>
>
>
> Compared to the text module (where deactivating JailNeurons yields a large increase in refusal probability), the increase here is much smaller. This suggests that:
>
>
>
> - While there do exist sparse visual neurons whose modification can nudge outputs toward safer responses,
>
> - the _primary_ locus of safety‑related and “moral” concepts resides in the text decoder, and the jailbreak signal carried purely by the vision module is relatively weaker.
>
> We will incorporate this multimodal analysis in the revised version to explicitly address the reviewer’s question and to better articulate the division of labor between visual and textual pathways during jailbreak.

---

> ### Author Response · Authors · 2025-11-21
> **Author Response (part 2 of 3)**
>
> ## 3. On neuron stability across architectures and finetuning checkpoints
>
>
> **Our response.**
>
> We thank the reviewer for this insightful question about stability, which is important for understanding how JailNeurons behave under model updates. We analyze stability from two perspectives: across architectures and across checkpoints within the same architecture.
>
> ### (i) Across different architectures
>
> Directly comparing neuron sets across architectures (e.g., Qwen‑VL‑8B vs. Janus‑Pro‑8B) is non‑trivial due to differences in layer counts, hidden sizes (e.g., 4096 vs. 3894 neurons per decoder layer), and training recipes. Neuron indices are not aligned and do not correspond to clearly comparable semantic functions.
>
> We observe the number of JailNeurons for different architectures. We find that different model architectures and training recipes can lead to varying quantities (proportions) of jailNeurons. For LLaVA and minigpt4, jailNeurons account for approximately 1% to 2%; whereas for Qwen-VL and janus-pro, the proportion is less than 1%.
>
> ### (ii) Within the same architecture under finetuning
>
> To study checkpoint‑level stability, we focus on three variants of LLaVA‑NeXT‑8B:
>
>
> 1.  **O_llava**: the official base model;
>
> 2.  **SS_llava**: a LoRA‑finetuned variant on ScreenSpots (grounding task);
>
> 3.  **FS_llava**: a LoRA‑safety‑tuned variant on FigStep (safety alignment).
>
> We first examine the proportion of JailNeurons in each model. Values denote the **percentage** of JailNeurons among all neurons in each layer:
>
>
> |masked_layer_id |1 |5|9|13|17|21|25|
> | :----: |:----: |:----: |:----: |:----: |:----: |:----: |:----: |
> |O_llava|1.81|1.76|1.42|1.76|1.22|0.93|1.22|
> |SS_llava|1.61|1.66|1.71|1.76|1.22|1.03|1.22|
> |FS_llava|0.92|0.78|0.78|0.68|0.63|0.53|0.73|
>
>
>
> We observe:
>
> - For **SS_llava** (fine‑tuned on a non‑safety task), the proportion of JailNeurons remains very close to that of **O_llava** across layers, suggesting that task‑oriented finetuning outside safety alignment has limited impact on the presence of JailNeurons.
>
> - For **FS_llava** (safety‑aligned), the proportion of JailNeurons decreases notably, consistent with the intuition that stronger built‑in safety reduces the number of neurons that actively support unsafe behavior.
>
>
> To further quantify stability, we measure how many JailNeurons in the fine‑tuned models are shared with the base model **O_llava**. Values denote the fraction of JailNeurons in each fine‑tuned model that are also JailNeurons in **O_llava**:
>
>
> |Overlap ratio|1 |5|9|13|17|21|25|
> | :----: |:----: |:----: |:----: |:----: |:----: |:----: |:----: |
> |SS_llava in O_llava|0.858|0.900|0.806|0.937|0.809|0.805|0.904|
> |FS_llava in O_llava|0.789|1.000|1.000|0.928|0.909|0.866|0.792|
>
>
> We find that:
>
>
> - For **SS_llava**, over 80% of its JailNeurons are inherited from **O_llava** across layers, indicating high stability under task‑oriented finetuning that does not target safety.
>
> - For **FS_llava**, the overlap remains very high (often close to or equal to 1.0), while the _total_ number of JailNeurons is reduced (as shown in the previous table). This suggests that safety alignment primarily prunes or suppresses a subset of existing JailNeurons, rather than creating a large number of new ones.
>
>
> Taken together, these results indicate that:
>
> - For non‑safety finetuning, the JailNeurons are largely preserved and remain stable.
>
> - For safety‑oriented finetuning, the JailNeurons in the safety‑aligned model are largely a subset of those in the original model, with relatively few new JailNeurons (< 20%) introduced.
>
> We believe this analysis provides an initial, quantitative characterization of JailNeuron stability across checkpoints.
>
> ## 4. On the limitation of white‑box settings and proxy‑based extensions
>
> **Our response.**
>
> We appreciate this thoughtful comment on deployment realism and broader impact. Our current method indeed assumes white‑box access. However, we note that such access is realistic in several common scenarios, including:
>
> - organizations deploying LVLMs in‑house or on‑premise;
>
> - research and open‑source communities working with publicly released checkpoints.
>
> These are also **typical assumptions** in prior work on neuron‑level safety analysis for LLMs and LVLMs [1][2].
>
> We agree that extending JDJN to more restricted settings is an important direction. We disscuss one realistic senario: currently, many specialized LVLMs are actually derived from open-source LVLMs after supervised fine-tuning (for example, there are many specialized models based on LLaVA available on Hugging Face). Our previous experiments find that for tasks outside of safety alignment, the JailNeurons in the fine-tuned models are almost identical to those in the pre-fine-tuned models. Therefore, the JDJN trained on open-source models (which allow for white-box access) could potentially serve as a proxy-based detector to act as a safety wrapper around the black-box model.

---

> ### Author Response · Authors · 2025-11-21
> **Author Response (part 3 of 3)**
>
> ## 5. On the narrow scope of benign distributions
>
> **Our response.**
>
> We appreciate the reviewer’s suggestion to broaden benign coverage, which indeed helps to more thoroughly assess false positives and over‑sensitivity. In response, we extend our evaluation to include two additional benign datasets:
>
>
>
> -  **ScreenSpots**: assesses LVLM performance on grounding tasks [3];
>
> -  **AndroidControl**: assesses LVLM performance on GUI agent control tasks [4].
>
>
>
> These datasets cover different benign usage patterns and interaction styles, thereby offering a more diverse set of non‑jailbroken inputs.
>
>
>
> We evaluate JDJN$_1$​ (trained on JailBreak‑V and MM‑Vet) on these new benign datasets. The false positive rates (FPR) are as follows:
>
>
>
> |Model|MM‑Vet|MM‑Bench|Normal|ScreenSpots|AndroidControl|
> | :----: |:----: |:----: |:----: |:----: |:----: |
> | LLaVA |0.000|0.000|0.019|0.022|0.012|
> |Janus‑Pro|0.000|0.000|0.044|0.002|0.033|
>
>
>
> Across all five benign datasets, JDJN maintains low FPR (≤ 0.05). This expanded evaluation supports the stability of JDJN$_1$​ under broader benign distributions and suggests good potential for realistic deployment scenarios where benign inputs are diverse and task‑specific.
>
> ------
>
> **Once again, we thank the reviewer for the insightful comments and for highlighting both the strengths and areas for improvement. We believe the suggested clarifications and new experiments substantially strengthen the manuscript.**
>
> [1] HiddenDetect: Detecting Jailbreak Attacks against Large Vision-Language Models via Monitoring Hidden States
>
> [2] How Alignment and Jailbreak Work: Explain LLM Safety through Intermediate Hidden States
>
> [3] SeeClick: Harnessing GUI Grounding for Advanced Visual GUI Agents
>
> [4] On the Effects of Data Scale on UI Control Agents

---

> > ### Comment · Reviewer_5qV5 · 2025-11-25
> > **Response to authors**
> >
> > Thanks for your detailed rebuttal. Most of my earlier concerns have been addressed. Regarding the overall novelty and contributions of the paper, I will maintain my original score.

---

> > > ### Author Response · Authors · 2025-11-25
> > >
> > > Thank you very much for your recognition of our work. If you have any further questions, please feel free to let us know. We appreciate your suggestions and are happy to answer your inquiries.

---

### Author Response · Authors · 2025-11-21
**Response to all reviewers**

We sincerely thank all reviewers for their thorough reviews and constructive feedback. We are encouraged that the reviewers recognize the innovations in our work, particularly the originality in conceptualization of **JailNeurons**, and the high efficiency and generalization of **JDJN**.

**Key Improvements and Clarifications**
1. Richer baselines and benchmarks.
- Adding three jailbreak defense methods;
- Adding three neuron-digging methods;
- Adding two benign datasets for LVLM;
- Adding two datasets for testing the phenomenon of model over-rejection.

2. A deeper study of the properties of JailNeurons.
- Causal relationship between JailNeurons and jailbreak behavior;
- Transferability of JailNeurons across different training datasets;
- Transferability of JailNeurons across different model architectures and checkpoints;
- The necessity of JailNeurons.

3. Clearer experimental settings.
- Specific configurations of baselines;
- Evaluation metrics;
- Criteria for determining the success of adversarial sample attacks.

4. More discussion and analysis of the experiments.
- Cost analysis of training the detector;
- Analysis of failure cases;
- Extension to access-limited scenarios.
- Probing the role of JailNeurons in vision modules.

Please see our detailed responses below each review and the blue-highlighted sections in the updated paper. We appreciate your feedback and welcome any additional questions.

---

### Meta-Review · Area_Chair_nASi · 2026-01-04

**Summary:**

This paper introduces JDJN, a novel jailbreak detection method for vision-language models that identifies a sparse set of "JailNeurons" linked to harmful outputs. The core innovation, the "sure‑to‑sorry" localization framework, is praised by reviewers for its originality and mathematically sound optimization. The resulting lightweight detector is both efficient and highly practical, as it requires no model modifications and demonstrates strong performance and generalization across new threats and data sources.

Overall, this paper is borderline but its strengths outweigh its weaknesses. The authors are strongly encouraged to integrate the discussion points raised during review into the final version.

**Reviewer Concerns:**

Overall, reviewers raised concerns across three main areas:
- Clarifying whether the identified neurons have a true causal role in jailbreak behavior, as opposed to being surface-level correlates.
- Questions regarding the necessity of the core masking component, potential overfitting to existing benchmarks, generalization to diverse benign distributions, and the completeness of baseline comparisons.
- Clarifying efficiency, real-world applicability in white- vs. black-box settings, and the precision of figures, metrics, and claims.

The authors have provided substantial additional experiments and explanations, which adequately address most of the reviewers' concerns.

**Reviewer Scores:**

This paper received four reviews, initially split between two borderline accept and two borderline reject recommendations. During the discussion phase, one of the dissenting reviewers changed their assessment to borderline accept, resulting in a final consensus of three borderline accepts. The other dissenting reviewer did not participate in the discussion.

After reviewing the authors' response, the concerns raised by the sole remaining critical reviewer, such as the necessity of the mask component and missing baseline comparisons, have been adequately addressed.

---

### Decision · Program_Chairs · 2026-01-26

Accept (Poster)